# DYNAMIC SEARCH FOR INFERENCE-TIME ALIGNMENT IN DIFFUSION MODELS

## ABSTRACT

Diffusion models have shown promising generative capabilities across diverse domains, yet aligning their outputs with desired reward functions remains a challenge, particularly in cases where reward functions are non-differentiable. Some gradient-free guidance methods have been developed, but they often struggle to achieve optimal inference-time alignment. In this work, we newly frame inference-time alignment in diffusion as a search problem and propose Dynamic Search for Diffusion (DSearch), which subsamples from denoising processes and approximates intermediate node rewards. It also dynamically adjusts beam width and tree expansion to efficiently explore high-reward generations. To refine intermediate decisions, DSearch incorporates adaptive scheduling based on noise levels and a lookahead heuristic function. We validate DSearch across multiple domains, including biological sequence design, molecular optimization, and image generation, demonstrating superior reward optimization compared to existing approaches.

## 1 INTRODUCTION

Diffusion models (Sohl-Dickstein et al., 2015; Ho et al., 2020; Song et al., 2020) have emerged as a powerful generative framework for a wide range of domains, from image synthesis to molecular design. While diffusion models excel at capturing complex data distributions, there is often a need to further optimize downstream reward functions, a task known as alignment. For instance, in image synthesis, we may seek to optimize rewards such as aesthetic scores. In drug design, the goal might be to optimize binding affinity.

Diffusion models can be adapted to maximize rewards. This alignment problem has been addressed by guiding generation at inference time using rewards. Classifier guidance (Dhariwal & Nichol, 2021) provides a standard scheme for doing this using the gradient of the reward functions, but critically depends on differentiable reward functions—an assumption that fails in many real-world scientific applications. In these domains, rewards are often non-differentiable or given in a black-box manner. For example, widely used docking softwares AutoDock Vina (Trott & Olson, 2010) for predicting binding affinity, which relies on physical simulations, as well as rewards derived from secondary structure estimation algorithms like DSSP (Kabsch & Sander, 1983) or structure predictors like AlphaFold3 (Abramson et al., 2024), which incorporate scientific knowledge via lookup tables, do not support gradient computation. Similarly, rewards based on widely-used molecular descriptors such as molecular fingerprints (Todeschini & Consonni, 2008) are inherently non-differentiable. Therefore, it is extremely difficult or infeasible to learn accurate differentiable surrogates for these scientific rewards. As a result, gradient-free guidance methods have gained increasing attention (Wu et al., 2024; Li et al., 2024). While proven simple and effective, they do not provide optimally accurate inference alignment. More sophisticated methods in this direction have yet to be explored.

In this work, we propose a novel gradient-free inference-time alignment method based on our new insight: framing inference-time alignment in diffusion models as a search problem. Pre-trained diffusion models inherently induce a tree structure that characterizes the generation process. By appropriately defining the search tree, search algorithms can be applied to maximize rewards effectively. Given the success of search in biochemical designs (Yang et al., 2017; Kajita et al., 2020; Yang et al., 2020; Swanson et al., 2024) and language models (Yao et al., 2024; Besta et al., 2024) for maximizing rewards in general, we believe search methods integrated into diffusion models would offer considerable potential for inference-time alignment. Specifically, we first establish the search tree formulation by subsampling from denoising processes of pre-trained diffusion models, assigning rewards to the leaf nodes, and introducing a heuristic function to evaluate intermediate nodes. Then, we propose "Dynamic Search for Diffusion (DSearch)" for inference-time alignment in

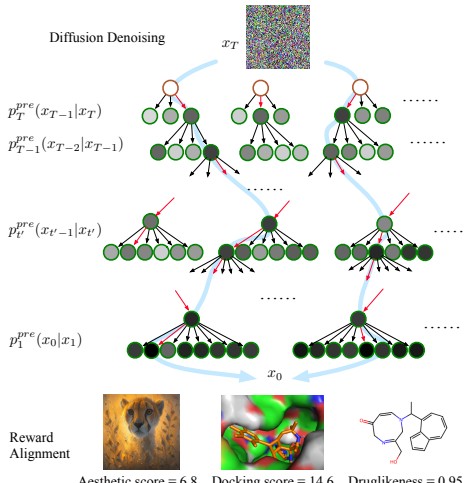

Diffusion Denoising

$x_T$

$p_T^{pre}(x_{T-1}|x_T)$

$p_{T-1}^{pre}(x_{T-2}|x_{T-1})$

$p_{t'}^{pre}(x_{t'-1}|x_{t'})$

$p_1^{pre}(x_0|x_1)$

$x_0$

Reward
Alignment

Aesthetic score = 6.8    Docking score = 14.6    Druglikeness = 0.95

Figure 1: **Inference-time alignment of diffusion model as a search problem.** We propose a dynamic search to maximize rewards efficiently and effectively. The top-down process visualizes the diffusion denoising trajectory starting from Gaussian noise down to the final sample $x_0$. Green circles indicate tree nodes, representing candidate samples at a time step, while darker nodes mark higher potential rewards. Red slashes denote selections, while nodes without selected children are pruned branches (suboptimal candidates eliminated during search). Blue arrows trace the final high-reward trajectory dynamically selected to maximize the downstream reward under computational budgets.

diffusion models. DSearch applies dynamic beam search which dynamically adjusts the beam size and tree width across time steps, as shown in Figure 1; in contrast, the naïve approach such as static beam search can lead to wasted computational resources when encountering suboptimal samples at intermediate steps.

Our contributions are summarized as follows. In brief, we propose a novel search framework for inference-time alignment in diffusion models. Specifically, we introduce a method, DSearch, which features dynamically reducing the beam width while extending the tree width. Meanwhile, DSearch incorporates a dynamic scheduling of tree expansion based on noise levels and a novel lookahead heuristic function for intermediate nodes, which further enhance the efficiency and guidance precision. We experimentally validate the effectiveness of our proposal across multiple domains, including biological sequence design, molecular structure optimization, and image generation. DSearch demonstrates strong reward optimization for generative tasks with balanced sample naturalness, diversity, and efficiency, making it particularly suitable for real-world applications.

## 2 PRELIMINARY

In this section, we introduce diffusion models and outline our objective of inference-time alignment.

### 2.1 DIFFUSION MODELS

Diffusion models (Sohl-Dickstein et al., 2015; Ho et al., 2020; Song et al., 2020) aim to learn a sampler $p^{\text{pre}}(\cdot) \in \Delta(\mathcal{X})$ over a given design space $\mathcal{X}$ (e.g., Euclidean space or discrete space) from a data distribution. The primary objective in training diffusion models is to establish a sequential mapping, i.e., a denoising process, that transforms from a noise distribution to the true data distribution. The training procedure follows several steps. First, a forward noising process $q_t : \mathcal{X} \to \Delta(\mathcal{X})$ is predefined, evolving over time from $t = 0$ to $t = T$. This noising process is often referred to as a policy, drawing from reinforcement learning terminology. The goal is then to learn a reverse denoising process $p_t$, where each $p_t : \mathcal{X} \to \Delta(\mathcal{X})$ ensures that the marginal distributions induced by the forward and backward processes remain equivalent.

Next, we explain how to obtain such $p_t$. For this purpose, we define the forward noising processes. When $\mathcal{X}$ is a Euclidean space, we typically use the Gaussian distribution $q_t(\cdot \mid x_t) = \mathcal{N}(\sqrt{\alpha_t}x_t, (1 - \alpha_t)\mathrm{I})$ as the forward noising process where $\alpha_t \in \mathbb{R}$ denote a noise schedule. Then, the backward process $p_t(\cdot|x_t)$ is parameterized as a normal distribution with mean

$$\frac{\sqrt{\alpha_t}(1 - \bar{\alpha}_{t-1})x_t + \sqrt{\bar{\alpha}_{t-1}}(1 - \alpha_t)\hat{x}_0(x_t; \theta)}{1 - \bar{\alpha}_t},$$

where $\bar{\alpha}_t = \prod_{i=1}^t \alpha_i$. Importantly, $\hat{x}_0(x_t)$ is treated as a predictor for $\mathbb{E}[x_0 \mid x_t]$.

*Remark* 2.1 (Parametrization). Note that alternative parametrizations, such as noise or scores, can also be used in place of $\hat{x}_0(x_t)$ (Luo, 2022).

### 2.2 INFERENCE-TIME ALIGNMENT

Our objective is to obtain natural designs that exhibit a high likelihood $p^{\text{pre}}(\cdot)$ while maximizing the reward $r : \mathcal{X} \to \mathbb{R}$.

This goal can be formulated as sampling from:

$$p^{(\alpha)}(\cdot) \propto \exp(r(x)/\alpha)p^{\mathrm{pre}}(\cdot). \tag{1}$$

Here, $\alpha$ is the temperature parameter, which is set low in practice, as our primary focus is optimizing rewards. Note this objective has been widely adopted in the context of alignment in generative models, including autoregressive models (Wang et al., 2024).

Many inference-time alignment techniques have also been proposed in diffusion models, which organically combine $\{p_t^{\mathrm{pre}}(\cdot \mid x_{t-1})\}$ and $r$. As shown in (Uehara et al., 2024b, Theorem 1), this goal is achieved by sampling from the following policy from $t = T$ to $t = 0$

$$p_{t-1}^{\star}(\cdot|x_{t-1}) \propto \exp(v_{t-1}(\cdot)/\alpha)p_{t-1}^{\mathrm{pre}}(\cdot|x_{t-1}). \tag{2}$$

Here, $v_{t-1}(\cdot)$ is soft value function defined as $v_{t-1}(\cdot) := \alpha \log \mathbb{E}_{x_0 \sim p^{\mathrm{pre}}(x_0|x_{t-1})}[\exp(r(x_0)/\alpha)|x_t]$, where the expectation is taken with respect to the distribution from the pre-trained policies. This soft value function acts as a look-ahead function that predicts future rewards from intermediate states. However, exact sampling from this policy $p_{t-1}^{\star}$ is not feasible since the soft value functions are unknown, and computing the normalizing constant is challenging due to the large action space. To address these challenges, several approaches, such as gradient-based classifier guidance or gradient-free guidance, have been proposed (refer to Section 5). While these methods have shown success, in this work, we introduce a more efficient search framework that extends beyond these approaches.

## 3 SEARCH FRAMEWORK FOR INFERENCE-TIME ALIGNMENT IN DIFFUSION

We aim to introduce an efficient search method for alignment in diffusion models. To this end, we define a formulation of search tree framework leveraging pre-trained diffusion models in this section.

We begin by examining the naïve approach to leverage pre-trained diffusion models. This involves defining a tree where each child is recursively determined by the support of the pre-trained diffusion models: $t \in [T]; \mathrm{Ch}(x_t) = \{x_{t-1} : p^{\mathrm{pre}}(x_{t-1}|x_t) > 0\}$. Then, the leaf nodes correspond to $\mathrm{Supp}(p^{\mathrm{pre}}) := \{x : p^{\mathrm{pre}}(x) > 0\}$. The alignment problem is then addressed by selecting the maximum (or top several) samples from the leaf nodes based on rewards, as this corresponds to: $\mathrm{argmax}_{x \in \mathrm{Supp}(p^{\mathrm{pre}})} r(x)$, which is equivalent to our goal in (1) with $\alpha = 0$. However, in practice, exact search within this tree is not feasible, as the tree's size is $O(|\mathcal{X}|^T)$ in the worst case. We proceed by explaining how to resolve this issue.

### 3.1 LIMIT TREE WIDTH: PRUNING WITH PRE-TRAINED POLICIES

Instead of using the entire support $\mathrm{Supp}(p^{\mathrm{pre}})$, we employ its empirical distribution. In the context of search, this involves constraining the tree width by sampling nodes from the pre-trained model during expansion, thereby limiting further growth to a specified threshold $w : [T] \to \mathbf{N}$. Specifically, the tree is recursively defined by setting child nodes

$$\mathrm{Ch}(x_t) = \{x_{t-1}[i]\}_{t=1}^{w(t)}, \ \{x_{t-1}[i]\}_{i=1}^{w(t)} \sim p^{\mathrm{pre}}(\cdot|x_{t-1}),$$

as illustrated in Figure 1. After defining this tree, the alignment problem is addressed by selecting leaf nodes with high rewards. Notably, when $w(t) = 1$ for all $t \in [T]$, this reduces to best-of-N sampling.

However, this approach still remains computationally expensive once the width exceeds 1, as the tree size grows to $O(w^T)$ where $w := \max_t w(t)$. One potential solution to this issue is to use heuristic functions that guide the search in intermediate nodes, avoiding the need to traverse the entire tree. Next, we introduce such heuristic functions.

### 3.2 DEFINE "HEURISTIC FUNCTIONS" IN NODES

We propose using "estimated" value functions as heuristic functions. The rationale is as follows. Suppose we take a greedy action at $x_{t-1}$ based on the exact value function. In this case, the decision simplifies to: $\mathrm{argmax}_{x \in \mathrm{Ch}(x_{t-1})} v_{t-1}(x)$, which corresponds to the soft optimal policy in equation (2) as $\alpha$ approaches 0, with pre-trained policies replaced by empirical distributions. While the remaining challenge is how to estimate such value functions, building on recent works, we introduce our novel approach in Section 3.3.

Figure 2: Illustration of DSearch. Our proposed dynamic search has expanding tree widths. We dynamically adjust weaker beams and reallocate their computational resources to other beams across time steps, fixing $w(t)b(t)$ while strategically scheduling $b(t)$.

### 3.3 LOOK-AHEAD HEURISTIC FUNCTION ESTIMATION

We also extend to construct more accurate estimations for value functions. The most commonly used approach in many contexts (e.g., DPS (Chung et al., 2022), reconstruction guidance (Ho et al., 2022), SVDD (Li et al., 2024)) is

$$\hat{v}_t(x_t) := r(\hat{x}_0(x_t)). \tag{3}$$

Intuitively, this is very natural since $\hat{x}_0(x_t)$ introduced in Section 2.1 is a one-step mapping from $x_t$ to $x_0$ (i.e., approximation of $\mathbb{E}[x_0 \mid x_t]$). Mathematically, this is based on the reasoning below. Recall that the definition of soft value functions involves an expectation w.r.t. $p_0^{\text{pre}}(\cdot|x_t)$. Then (3) is derived by replacing the probability $p_0^{\text{pre}}(\cdot|x_t)$ with its mean:

$$p_0^{\text{pre}}(\cdot|x_t) \underbrace{\approx}_{(A)} \delta(\mathbb{E}[x_0|x_t]) \underbrace{\approx}_{(B)} \delta(\hat{x}_0(x_t)). \tag{4}$$

While this approximation has been widely used due to its training-free nature, we propose using a more accurate approach.

---

**Algorithm 1** Look-Ahead Search for Value Estimation
---

1: **Require**: Lookahead step $K$, duplication size $M$
2: $\{x_{t-K}^{\langle s \rangle}\}_{s=1}^M \sim p_{t-K}^{\text{pre}}(x_{t-K}|x_t)$.
3: **Output**: $1/M \sum_{s=1}^M r(\hat{x}_0(x_{t-K}^{\langle s \rangle}))$

---

The look-ahead value estimation is summarized in Algorithm 1. It consists of three steps: running $M$ particles for $K$ steps ahead (Line 2), mapping to $r(x_0)$ using $\hat{x}_0(\cdot)$, and evaluate its reward. Our approach is based on the following approximation:

$$p_0^{\text{pre}}(\cdot|x_t) = \mathbb{E}_{p_{t-k}^{\text{pre}}(x_{t-k}|x_t)}[p_0^{\text{pre}}(\cdot|x_{t-k})] \approx 1/M \sum_s \delta(\mathbb{E}[x_0|x_{t-k}^{\langle s \rangle}]) \approx 1/M \sum_s \delta(\hat{x}_0(x_{t-k})).$$

Now we compare this with the approximation used in the existing method (4). Here, the approximation in (A) of (4) is enhanced by considering multiple particles. Meanwhile, the approximation in (B) of (4) is improved, as $\hat{x}_0(x_t)$ is expected to become more accurate as $t$ approaches 0. From the next section, assuming we have a reliable estimate $\hat{v} : \mathcal{X} \to \mathbb{R}$, we present our proposed search algorithm.

## 4 DYNAMIC SEARCH FOR DIFFUSION

In this section, we present our proposed search algorithm, Dynamic Search for Diffusion (DSearch), for inference-time alignment in diffusion models.

### 4.1 DYNAMIC SEARCH TREE EXPANSION

Based on the tree formulation in Section 3, a straightforward yet effective approach is to perform beam search with a fixed tree width and beam size, guided by heuristic functions. However, the underlying challenge is computational efficiency, as static tree search may lead to wasted computational resources when encountering suboptimal samples at intermediate steps. To address this issue, we adopt a dynamic strategy for tree search.

We propose a dynamic search algorithm with expanding tree width that dynamically adjusts the beam size and tree width across time steps by beam schedule $b(\cdot) : [T] \to \mathbf{N}$ and tree schedule

---

**Algorithm 2** Dynamic Search for Diffusion (DSearch)

---

1: **Require**: Heuristic functions $\{\hat{v}_t\}_{t=T}^0$ (refer to Section 3.3), Search set $\mathcal{A}$, (monotonically decreasing) beam width $b(\cdot) : [T] \to \mathbf{N}$, tree width $w(\cdot) : [T] \to \mathbf{N}$
2: **for** $t \in [T+1, \cdots, 1]$ **do**
3:     **if** $t \in \mathcal{A}$ **then**
4:         For each beam $j \in [b(t)]$, we expand the node as $\mathrm{Ch}(x_t^{(j)}) = \{x_{t-1}^{(j)}[i]\}_{i=1}^{w(t)} \sim p^{\mathrm{pre}}(\cdot|x_t^{(j)})$ and perform greedy selection

$$z_{t-1}^{(j)} = \mathrm{argmax}_{x \in \mathrm{Ch}(x_t^{(j)})} \hat{v}(x)$$

5:         Change beam width from $b(t)$ to $b(t-1)$, i.e., set

$$\{x_{t-1}^{(j)}\}_{j=1}^{b(t-1)} := \mathrm{Selection}(\{z_{t-1}^{(j)}\}_{j=1}^{b(t)})$$

        where $\mathrm{Selection}(\cdot)$ is a function choosing top $b(t-1)$ samples with $\hat{v}(\cdot)$ among $\{z_{t-1}^{(j)}\}_{j=1}^{b(t)}$.
6:     **else**
7:         Set $\mathrm{Ch}(x_t^{(j)}) = x_{t-1}^{(j)} \sim p^{\mathrm{pre}}(\cdot|x_t^{(j)})$
8:     **end if**
9: **end for**
10: **Output**: $\{x_0^{[j]}\}$

---

$w(\cdot) : [T] \to \mathbf{N}$, which significantly outperforms static beam search methods. A practical question is how to control the dynamic beam size and tree width. Given the allocated memory budget during inference, we typically select these values under the constraint $w(t)b(t) = C$, where $C$ is a constant. Our design for tree expansion with dynamic beam-tree width is outlined in Algorithm 2. Intuitively, if a beam performs poorly, we apply early stopping for that beam and allocate its computational resources to other beams by increasing the tree width, as illustrated in Figure 2. This step is executed in Line 4 of our algorithm. Note that the set $\mathcal{A}$ in line 3 is determined by search scheduling, which is detailed in Section 4.2 below. Since the tree width $w(t)$ is determined by $C/b(t)$, we focus primarily on the selection of the beam width below.

Here we introduce beam scheduling technique, which aims to improve sample selection by initially over-sampling a larger batch of candidates and then progressively pruning weaker samples at intermediate steps. Instead of treating all samples equally throughout the entire diffusion process, this approach selectively retains high-quality candidates, allowing computational resources to be focused on the most promising sequences. Given an initial beam size $b(0)$ and the final beam size $b(T)$, we can apply exponential scheduling, which is an interpolation following $b(t) = b(0) \cdot \left(\frac{b(T)}{b(0)}\right)^{t/T}$, and illustrated as the left histogram (brown) of Figure 2. Exponential beam scheduling is particularly effective, as it ensures that early-stage candidates are explored broadly while later-stage refinement is performed on only the most promising samples. Note that while we generally recommend the exponential way, we consider the beam scheduling strategy as a hyperparameter and experiment with multiple functions, which is detailed in Appendix F.3.1.

## 4.2 SCHEDULING OF SEARCH NODES

In Algorithm 2, to efficiently allocate computational resources during diffusion inference, we propose using a time-aware scheduling mechanism to dynamically determine the expansion of the search tree (i.e., Line 3). Hereafter, we explain its details.

We first start with the intuition on why we need such a scheduling mechanism. Unlike in autoregressive models (Feng et al., 2023; Hao et al., 2023), where the importance of each step remains relatively uniform, diffusion decoding exhibits sparse information in early steps and increasingly dense information as time approaches the final stages. Also, when $t$ is large, heuristic functions are typically less accurate due to the high noise in the state at early times. This phenomenon motivates a scheduling strategy to focus search efforts where it is most impactful, particularly in later time steps, thereby balancing computational efficiency and model performance. Specifically, we define the set $\mathcal{A} \subset [T]$, which corresponds to the nodes selected for expansion as follows. Note the computational time is reduced from $O(TC)$ to $O(|\mathcal{A}|C)$, which equals to $O(T\bar{C})$, where $\bar{C} = (|\mathcal{A}|C + T - |\mathcal{A}|)/T$.

To define such a set $\mathcal{A}$. Given a budget for the size of $\mathcal{A}$ as $C^\dagger$, we consider the exponential scheduling function $\mathcal{A} = \{t \in [T] | \mathbb{1}(U_{(0,1)} \leq e^{\beta(T-t)/T}) = 1\}$, thus by integration $C^\dagger = T(1 - e - \beta/\beta)$, with $C^\dagger \approx T$ when $\beta \to 0$ (uniform inclusion) and $C^\dagger \approx 0$ as $\beta \to \infty$ (aggressive filtering). Thus we can control the total search based on computational preference. An illustration is as the left histogram (green) of Figure 2, where interstices (choice of minimal tree width) become less frequent as $t$ progresses to 0. Exponential search scheduling is generally effective, as prioritizing late-stage refinement leads to better optimization. Still, we consider the scheduling function as a hyperparameter and explore multiple cases, detailed in Appendix F.3.2.

## 5 RELATED WORKS

**Gradient-Free guidance in diffusion models.** We focus on inference-time methods for optimizing rewards in diffusion models without fine-tuning. The early approach generates multiple samples and select the top samples based on the reward functions, known as best-of-N in autoregressive models (Stiennon et al., 2020; Nakano et al., 2021; Touvron et al., 2023; Beirami et al., 2024; Gao et al., 2023). This approach is significantly less efficient, since merely interfering with the final state does not shift the overall distribution effectively. Recently, gradient-free methods have been proposed to guide generation with non-differentialble rewards at inference time. SMC (sequential monte carlo)-based methods (Wu et al., 2024; Trippe et al., 2022; Dou & Song, 2024; Phillips et al., 2024; Cardoso et al., 2023; Kim et al., 2025; Singhal et al., 2025; Ma et al., 2025; Guo et al., 2025) perform resampling with replacement to approximate a non-deteriorated optimal policy. While they are originally designed for conditioning (by setting rewards as classifiers), they can also be applied to reward maximization. The other approach is SVDD (Li et al., 2024)), which performs value-based importance sampling in an iterative nature using soft value functions, approximating sampling from the optimal policy. While these approaches are related, our proposed method is fundamentally different, where we frame the task as a search problem. From this perspective, we introduce a search algorithm with dynamically controlled beams, a technique not explored in existing work. One concurrent work (Singhal et al., 2025) studies inference-time scaling for diffusion; however, our contributions differ substantially in that we dig into adaptive search methods such as dynamic beam/tree scheduling proposals.

**Search and decoding in autoregressive models.** The decoding strategy, which dictates how sentences are generated from the model, is a critical component of text generation in autoregressive language models (Wu et al., 2016; Chorowski & Jaitly, 2016; Leblond et al., 2021). Recent studies have explored inference-time techniques for optimizing downstream reward functions (Dathathri et al., 2019; Yang & Klein, 2021; Qin et al., 2022; Mudgal et al., 2023; Zhao et al., 2024; Han et al., 2024). Search algorithms, such as Monte Carlo Tree Search (MCTS) (Kocsis & Szepesvári, 2006; Browne et al., 2012; Hubert et al., 2021; Xiao et al., 2019), have also been explored in decoding for autoregressive models. More recently, several studies (Yao et al., 2024; Besta et al., 2024) showed the potential of applying search to LLMs for enhancing performances on text-based reasoning tasks. Others have applied MCTS to improve the performance of LMs (Xie et al., 2024; Chen et al., 2024; Zhang et al., 2024; Zhou et al., 2024; Hao et al., 2023) on math benchmarks (Cobbe et al., 2021) or synthetic tasks (Yao et al., 2022; Valmeekam et al., 2023). However, such sophisticated search methodology in decoding is largely under-explored in diffusion models.

For more related works on fine-tuning and gradient-based methods, please refer to Appendix B.

## 6 EXPERIMENTS

We conduct experiments to assess the performance of our algorithm relative to baselines and its sensitivity to hyperparameters. We start by outlining the experimental setup, including baselines and tasks, and then present the results. The code is available at this anonymous link.

### 6.1 EXPERIMENTAL SETUP

**Baselines.** We compare DSearch to several representative methods performing reward maximization during inference. The **pre-trained** baseline generates samples using pre-trained diffusion models. **Best-of-N** generates samples from pre-trained models and select the top $1/N$ samples. **DPS (Chung et al., 2022)** is a widely used training-free version of classifier guidance. For discrete diffusion, we combine it with the state-of-the-art approach (Nisonoff et al., 2024). **SMC** resamples among batch samples at each time step from the weighted distribution based on value estimations. **SVDD** performs value-based importance sampling with fixed duplication-size at each time step.

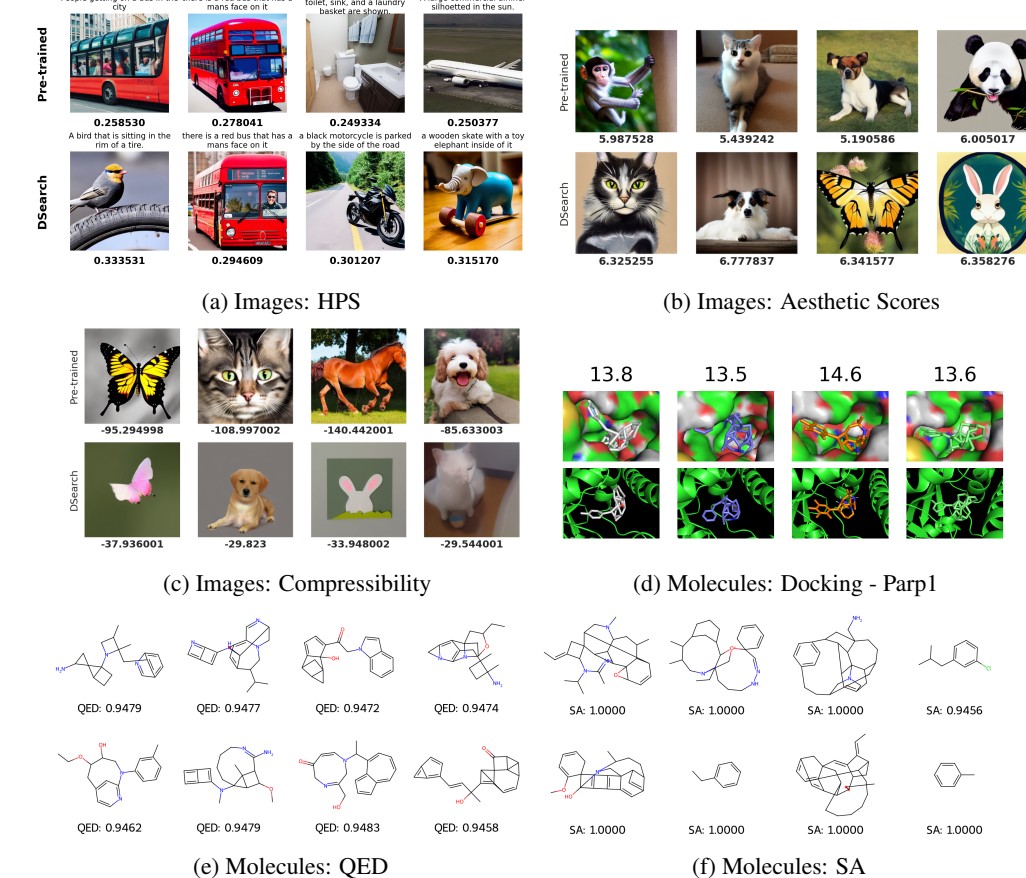

Figure 3: Generated samples from DSearch. For more samples, please refer to Appendix H.4. Note that the surfaces and ribbons in (e) are representations of the target proteins, while the generated small molecules are displayed in the center.

Hereafter, we explain more about how we compare with each algorithm. Since DSearch uses $\bar{C}$ times of computation compared to baseline sampling, we set $N = \bar{C}$ for Best-of-N and duplication-size $\bar{C}$ for SVDD, as well as use Best-of-N ($N = \bar{C}$) on top of DPS and SMC, to ensure that the computational budget during inference is approximately equivalent across different methods. Further details are provided in Appendix F.2. For DSearch, implementation details are provided in Appendix F.3. Unless otherwise stated, we use exponential search and beam scheduling.

**Tasks & Rewards.** We introduce the pre-trained diffusion models and downstream reward functions used. Further details are provided in Appendix F.1. For **images**, we use Stable Diffusion v1.5 as the pre-trained diffusion model ($T = 50$). For downstream reward oracles, we use compressibility, aesthetic score (LAION Aesthetic Predictor V2 in Schuhmann (2022)) and human preference score (HPS V2 in Wu et al. (2023)), as employed by Black et al. (2023); Fan et al. (2023). For **biological sequences**, we use the discrete diffusion model (Sahoo et al., 2024), trained on datasets from Gosai et al. (2023) for DNA enhancers, and those from Sample et al. (2019) for 5'Untranslated regions (5'UTRs), as our pre-trained diffusion model ($T = 128$). For the reward oracles, we use an Enformer model (Avsec et al., 2021) to predict activity for enhancers under cell-specificity, specifically in the HepG2 cell line. For 5'UTRs, we respectively use ConvGRU models to predict the mean ribosomal load (MRL) measured by polysome profiling (Sample et al., 2019), and the stability measured by half life (Agarwal & Kelley, 2022). Note that the stability reward is *non-differentiable* since the half life of 5'UTR is measured after concatenation with coding regions and 3'Untranslated regions. These tasks are highly relevant for cell and RNA therapies, respectively (Taskiran et al., 2024; Castillo-Hair & Seelig, 2021). For **molecules**, we use GDSS (Jo et al., 2022), trained on ZINC-250k (Irwin & Shoichet, 2005), as the pre-trained diffusion model ($T = 1000$). For reward oracles, we use drug-likeness (QED) and synthetic accessibility (SA) calculated by RDKit, as well as binding affinity to protein Parp1 (Yang et al., 2021) measured by docking score (DS) (calculated by QuickVina 2 (Alhossary et al., 2015)), which are all *non-differentiable feedbacks*. Here, we renormalize SA to $(10 - \text{SA})/9$ and docking score to $\max(-\text{DS}, 0)$, so that a higher value indicates better performance. These tasks are critical for drug discovery.

Table 1: Performance of different methods on alignment tasks w.r.t. reward, NLL/quality, and diversity. The computation budget $\bar{C}$ for the image compressibility, aesthetic and HPS tasks are 40, 45 and 55, Enhancer, 5'UTR MRL, and 5'UTR Stability tasks 100, 50, and 80, and molecular tasks 50, respectively. ↑ indicates higher values correspond to better performance while ↓ indicates lower for better. **bold** highlights the best performances.

| Method | Image Compressibility | | | Image Aesthetic Score | | | Image Human Preference Score | | |
|---|---|---|---|---|---|---|---|---|---|
| | Compressibility↑ | Quality↑ | Diversity↑ | Aesthetic↑ | Quality↓ | Diversity↑ | HPS↑ | Quality↓ | Diversity↑ |
| Pre-trained | -95.7 ± 7.8 | 11.4 ± 7.4 | 0.2852 ± 0.0301 | 5.45 ± 0.15 | 11.4 ± 7.4 | 0.2852 ± 0.0302 | 0.2729 ± 0.0037 | 14.5 ± 1.3 | 0.5161 ± 0.0476 |
| Best-N | -65.9 ± 3.4 | 24.0 ± 6.4 | 0.2972 ± 0.0283 | 6.25 ± 0.05 | 3.2 ± 2.3 | 0.2713 ± 0.0306 | 0.2907 ± 0.0006 | 12.1 ± 10.2 | 0.3182 ± 0.0322 |
| DPS | -61.0 ± 4.9 | 22.7 ± 1.3 | 0.2392 ± 0.0499 | 6.16 ± 0.07 | 6.1 ± 2.9 | 0.2875 ± 0.0184 | 0.2971 ± 0.0026 | 14.1 ± 3.5 | 0.4173 ± 0.0304 |
| SMC | -66.0 ± 7.8 | 21.9 ± 7.8 | 0.1825 ± 0.0791 | 6.08 ± 0.05 | 4.7 ± 0.8 | 0.0649 ± 0.0347 | 0.2771 ± 0.0015 | 17.6 ± 1.8 | 0.4445 ± 0.0230 |
| SVDD | -37.3 ± 6.6 | 46.7 ± 1.6 | 0.2758 ± 0.0363 | 6.37 ± 0.26 | 4.6 ± 5.7 | 0.2655 ± 0.0540 | 0.2970 ± 0.0051 | 22.1 ± 8.0 | 0.4577 ± 0.0144 |
| DSearch | **-35.7** ± 4.2 | 42.7 ± 0.9 | 0.3156 ± 0.0111 | **6.54** ± 0.12 | 5.8 ± 10.5 | 0.2667 ± 0.0166 | **0.3133** ± 0.0058 | 16.5 ± 4.6 | 0.4323 ± 0.0534 |

| Method | Enhancer HepG2 | | | 5'UTR MRL | | | 5'UTR Stability | | |
|---|---|---|---|---|---|---|---|---|---|
| | HepG2↑ | NLL↓ | Diversity↑ | MRL↑ | NLL↓ | Diversity↑ | Stability↑ | NLL↓ | Diversity↑ |
| Pre-trained | 0.305 ± 0.295 | 261.0 ± 0.5 | 0.7197 ± 0.1650 | 0.345 ± 0.112 | 68.4 ± 0.2 | 0.7380 ± 0.1263 | 0.212 ± 0.010 | 68.4 ± 0.3 | 0.7375 ± 0.1735 |
| Best-N | 3.319 ± 0.152 | 263.0 ± 0.8 | 0.7097 ± 0.1703 | 1.009 ± 0.006 | 68.0 ± 0.4 | 0.7280 ± 0.1248 | 0.342 ± 0.002 | 68.9 ± 0.2 | 0.7275 ± 0.1710 |
| DPS | 3.665 ± 0.222 | 258.0 ± 2.1 | 0.7454 ± 0.0755 | 0.995 ± 0.016 | 72.0 ± 0.2 | 0.7408 ± 0.0956 | 0.419 ± 0.002 | 67.0 ± 0.5 | 0.6040 ± 0.2188 |
| SMC | 5.601 ± 0.208 | 288.0 ± 1.0 | 0.5737 ± 0.3563 | 1.008 ± 0.013 | 68.5 ± 0.5 | 0.5544 ± 0.2857 | 0.329 ± 0.006 | 69.0 ± 0.6 | 0.4856 ± 0.4068 |
| SVDD | 7.040 ± 0.068 | 246.2 ± 5.3 | 0.7159 ± 0.1024 | 1.356 ± 0.009 | 66.7 ± 0.8 | 0.6349 ± 0.2027 | 0.469 ± 0.002 | 69.2 ± 0.8 | 0.7309 ± 0.1572 |
| DSearch | **7.245** ± 0.502 | 260.1 ± 1.9 | 0.7063 ± 0.1684 | **1.521** ± 0.011 | 68.6 ± 0.6 | 0.6258 ± 0.2135 | **0.533** ± 0.004 | 71.0 ± 0.7 | 0.7001 ± 0.1783 |

| Method | Molecule Drug-likeness | | | Molecule Synthetic Accessibility | | | Molecule Binding Affinity - Parp1 | | |
|---|---|---|---|---|---|---|---|---|---|
| | QED↑ | NLL↓ | Diversity↑ | SA↑ | NLL↓ | Diversity↑ | Docking Score↑ | NLL↓ | Diversity↑ |
| Pre-trained | 0.656 ± 0.007 | 958 ± 58 | 0.8733 ± 0.1580 | 0.652 ± 0.006 | 971 ± 69 | 0.8429 ± 0.2227 | 7.2 ± 0.5 | 971 ± 32 | 0.7784 ± 0.2998 |
| Best-N | 0.887 ± 0.008 | 943 ± 33 | 0.8779 ± 0.1579 | 0.921 ± 0.014 | 946 ± 62 | 0.8442 ± 0.2220 | 10.2 ± 0.4 | 951 ± 22 | 0.7938 ± 0.3052 |
| DPS | 0.885 ± 0.019 | 971 ± 41 | 0.8961 ± 0.0761 | 0.968 ± 0.026 | 917 ± 57 | 0.8968 ± 0.0752 | 11.6 ± 0.1 | 948 ± 63 | 0.8882 ± 0.0581 |
| SMC | 0.796 ± 0.007 | 1086 ± 21 | 0.6441 ± 0.2591 | 0.633 ± 0.007 | 1050 ± 28 | 0.6894 ± 0.2268 | 10.6 ± 0.5 | 957 ± 36 | 0.5092 ± 0.3673 |
| SVDD | 0.931 ± 0.003 | 1049 ± 24 | 0.8920 ± 0.0589 | 0.986 ± 0.019 | 1068 ± 24 | 0.8633 ± 0.2277 | 12.7 ± 0.2 | 993 ± 25 | 0.8980 ± 0.0635 |
| DSearch | **0.946** ± 0.002 | 911 ± 28 | 0.8424 ± 0.2195 | **1.000** ± 0.009 | 892 ± 61 | 0.8546 ± 0.2424 | **13.7** ± 0.3 | 731 ± 35 | 0.7650 ± 0.2934 |

(a) Image: Aesthetic  (b) Image: HPS  (c) 5'UTRs: stability  (d) Molecule: SA

Figure 4: Reward (median & standard deviation) under different constraints $\bar{C}$.

**Metrics.** We measure the target reward as well as naturalness and diversity metrics for comprehensive evaluation. We calculate the negative log-likelihood (NLL) of the generated samples w.r.t the pretrained model to measure how likely the samples are to be natural. The likelihood is calculated using the ELBO of the pretrained diffusion model. For images, we use BRISQUE to assess the quality (naturalness) of generated samples (Mittal et al., 2011). We also evaluate the diversity of generated samples. A higher diversity score indicates greater variability in generation, ensuring broader exploration of the data space. For discrete biological sequences, we measure diversity using the pairwise distance of one-hot representation subtracted by 1 to capture structural variations. For images, we use CLIP (Radford et al., 2021) embeddings of samples to calculate average pairwise cosine similarity. For molecules, we use Tanimoto similarity on molecular Morgan fingerprints (ECFP), with diversity quantified as the average pairwise similarity of generated molecules, subtracted by 1.

## 6.2 EFFECTIVENESS OF DSEARCH

We compare the performance of DSearch with other methods. The main results are summarized in Table 1 on page 8. To visualize the generated samples, we also present several examples in Figure 3. Further results and studies, including runtime, more metrics and ablations are in Appendices E,G,H.

DSearch achieves superior reward performance across all evaluated tasks, consistently outperforming baselines. This trend is particularly evident in biological sequence generation tasks, where DSearch exhibits significantly higher scores in HepG2 enhancer activity, 5'UTR MRL, and stability. The improvement over methods such as Best-of-N, SVDD, and SMC suggests that DSearch's dynamic tree search effectively prioritizes high-reward samples while maintaining efficient exploration. While DSearch generally improves sample rewards, its naturalness remains competitive with baselines. In molecular generation tasks, DSearch achieves lower NLL compared to baselines, suggesting that it generates chemically realistic molecules. DSearch also exhibits a balance between diversity and reward, ensuring a reasonable level of diversity while significantly enhancing reward. In contrast, the baseline SMC, which rely on batch resampling strategies, show a marked drop in diversity.

In Figure 4, we illustrate how DSearch performance scales with computational budget $\bar{C}$. As $\bar{C}$ increases, reward scores improve for all methods, but the gains are most pronounced for DSearch. This shows that dynamic tree search effectively utilizes additional computation to align samples.

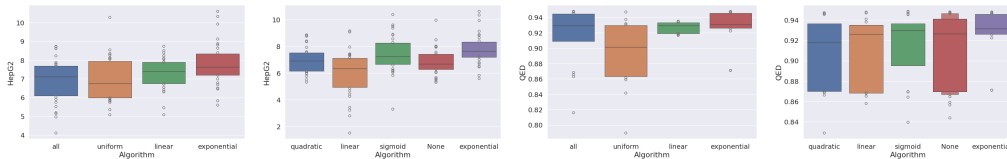

(a) Search Schedule (DNA)    (b) Beam Schedule (DNA)    (c) Search Schedule (Molecule)    (d) Beam Schedule (Molecule)

Figure 5: Reward distributions of generated samples using DSearch with different scheduling algorithms. We fix $\bar{C} = 40$ for DNA task and $\bar{C} = 20$ for molecular task. For search scheduling, "all" has $|\mathcal{A}| = T$ while other algorithms have $|\mathcal{A}|/T = 65\% \pm 1\%$. For beam scheduling, we use $\frac{b(T)}{b(0)} = 4$ for different algorithms except "None", which does not use beam reduction.

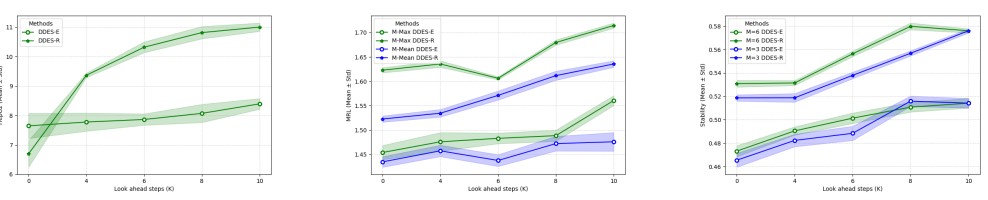

(a) Different $K$ ($M$-Max, $M$=6, DNA)    (b) Different pooling ($M$=6, 5'UTR MRL)    (c) Different $M$ ($M$-Max, 5'UTR stability)

Figure 6: Reward (median and standard deviation) of generated samples with different lookahead hyperparamters. We fix $\bar{C} = 40$.

### 6.3 EFFECTIVENESS OF SCHEDULING SEARCH EXPANSION

**Search scheduling.** We compare different search scheduling strategies, including uniform, linear, exponential, and no search schedules (detailed in Appendix F). As shown in Figure 5(a,c), we observe that exponential scheduling achieves better rewards while reducing computational cost by 35% compared to the no scheduling ("all") baseline. This suggests that focusing search efforts in the later steps of the generation process leads to better sample quality without requiring a proportional increase in computation. Linear and uniform scheduling also improve efficiency but do not reach the same level of performance, as they distribute search operations more evenly across time steps. These results validate that adaptive scheduling allows for significant computational savings while maintaining or even improving rewards, highlighting the importance of strategic search in diffusion.

**Beam scheduling.** We also evaluate different beam scheduling strategies, including quadratic, linear, sigmoid, exponential, and no pruning schedules. From Figure 5(b,d), we observe that dropping weaker samples through exponential beam scheduling performs the best. This demonstrates that reducing the search space aggressively in earlier steps allows for wider and more refined exploration later, adapting to the dynamic nature of the search. These results indicate that progressively focusing efforts on high-quality samples enhances overall alignment performance without increasing computational overhead, which is a key factor in DSearch.

### 6.4 EFFECTIVENESS OF LOOKAHEAD MECHANISM

Another component of DSearch is the lookahead mechanism introduced in Section 3.3, which strengthens the reward estimation of intermediate states. We explore the impact of different lookahead horizons $K$ and the number of forward evaluations $M$. For each sample, we generate $M$ lookaheads of $K$ steps, compute the corresponding final rewards, and select the best intermediate states either by the maximum or mean of these evaluations. From Figure 6, we observe that increasing $K$ consistently improves performance across different tasks, as it allows for a more informed selection of intermediate states. However, the gains saturate beyond a certain threshold, suggesting a limit of gain from the estimation accuracy. Additionally, choosing states by maximum reward generally outperforms averaging, as it ensures that the highest-quality rollouts guide the generation process. The effect of $M$ is more subtle; higher $M$ leads to better optimization in some tasks where exploration is crucial, such as 5'UTR stability.

## 7 CONCLUSION

This work builds on works in diffusion models, value-guided generation, and search algorithms, proposing a coherent framework for inference alignment. Our proposals open new avenues for tackling alignment tasks with diffusion models, a powerful tool for property-driven generation. Our studies show that DSearch effectively balances reward maximization and sample diversity, while maintaining reasonable likelihood.

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

## A    BROADER IMPACT

This paper presents work whose goal is to advance the field of Deep Learning, particularly diffusion models. While this research primarily contributes to technical advancements in generative modeling, it has potential implications in domains such as drug discovery and biomolecular engineering. We acknowledge that generative models, particularly those optimized for specific reward functions, could be misused if not carefully applied. However, our work is intended for general applications, and we emphasize the importance of responsible deployment and alignment with ethical guidelines in generative AI. Overall, our contributions align with the broader goal of machine learning methodologies, and we do not foresee any immediate ethical concerns beyond those generally associated with generative models.

## B    FURTHER RELATED WORKS

We further discuss works on diffusion post-training in a broader context.

**Fine-tuning of diffusion models.** Several methods exist for fine-tuning generative models to optimize downstream reward functions, such as classifier-free guidance (Ho & Salimans, 2022), RL-based fine-tuning (Fan et al., 2023; Black et al., 2023), and its variants (Dong et al., 2023; Wallace et al., 2024). However, these approaches come with caveats, including high computational costs and the risk of easily forgetting pre-trained models. In this work, we focus on inference-time techniques that eliminates the need for fine-tuning generative models.

**Gradient-based guidance in diffusion models.** Classifier guidance (Dhariwal & Nichol, 2021; Song et al., 2020) has been widely used to condition pre-trained diffusion models without fine-tuning. Although these methods do not originally focus on optimizing reward functions, they can be applied for this purpose (Uehara et al., 2024a). In this approach, an additional derivative of a certain value function is incorporated into the drift term (mean) of pre-trained diffusion models during inference. Subsequent variants (*e.g.*, Chung et al. (2022); Ho et al. (2022); Bansal et al. (2023); Guo et al. (2024); Wang et al. (2022); Yu et al. (2023); Nisonoff et al. (2024)) have been proposed to simplify the learning of value functions. However, these methods require the *differentiability* of proxy models, which limits their applicability to non-differentiable features/reward feedbacks commonly encountered in scientific domains. Additionally, this approach cannot be directly extended to discrete diffusion models (e.g., (Lou et al., 2023; Shi et al., 2024; Sahoo et al., 2024)) in a principle way. Note a notable exception of classifier guidance tailored to discrete diffusion models has been recently proposed by Nisonoff et al. (2024). However, our approach can be applied to both continuous and discrete diffusion models in a unified manner. Furthermore, their practical method requires the differentiability of proxy models.

## C    FAQ & DISCUSSIONS

In this section, we explore several additional questions on the design of DSearch.

### C.1    DOES DYNAMIC SEARCH REDUCE DIVERSITY?

Here we include in-depth analysis on how the DSearch design preserve diversity. In DSearch, we intentionally avoid redundant sampling from the same parent nodes by designing oversampling and controlled beam/tree scheduling. At the beginning of the reverse process, we oversample more initial beams than needed in the final output to enable broader exploration. At each denoising step, while we selectively retain high-reward beams and discard certain beams, for each retained beam, we select exactly one of its own next states as the node to continue with. Crucially, this ensures that no two beams at any timestep share the same parent. Each child has its unique parent beam, maintaining diversity across paths. We control the beam width to decrease under a certain schedule and thus tree width increases: the computational resources freed up by discarded beams are allocated to other beams. At the final step, we have the number of output samples (beams) we need, and each sample comes from a distinct ancestral trajectory. This design guarantees that the final samples are not collapsed to descendants of same parents or ancestors. Thus our dynamic search does not lead to near identical samples, but rather makes well use of resources by focusing the search on promising paths.

In addition, we include diversity metrics for all domains and tasks, including pairwise cosine for sequences and molecules, CLIP distance for images. Additional molecular metrics including various validity and diversity metrics are provided in Appendix G.5. Results show that DSearch maintains comparable or better diversity than SVDD and Best-of-N for the same reward level. This is due to oversampling and dynamic width control, which help maintain multiple diverse beams while still guiding toward high-reward modes. Meanwhile, a variant of DSearch introduced in Appendix E, DSearch-R (aggressive version) does reduce diversity as expected, which we discuss in Appendix E.

## C.2 Why is the Gradient-Free Design Beneficial?

our work specifically targets a common and important class of real-world problems where reward functions are non-differentiable, sparse, black-box, or expensive to compute. This includes many settings in scientific discovery domains.

- For drug design, we often want to optimize physics-based feedback such as Autodock Vienna (Trott & Olson, 2010) for binding affinity. Many powerful science models include non-differentiable features based on domain knowledge in our target scores. E.g., the famous Alphafold (Abramson et al., 2024) incorporates lots of non-differentiable features such as MSA (Lipman et al., 1989) and conformation search for functional scores. Another well-known non-differentiable feature is molecular fingerprints (Rogers & Hahn, 2010).
- In many scientific tasks, training a differentiable reward model is infeasible due to the lack of data. E.g., for many novel targets (new protein binding pockets), there is no known binder (molecule), let alone a sufficient labeled dataset to train a neural predictor.
- Even with data, we emphasize that it is extremely hard to construct differentiable models in these scenarios. Since practical calculation of many scientific properties requires chemical algorithms or physical simulations (as the examples above), which even state-of-the-art ML models cannot approximate with good accuracy, and are not scientifically trustworthy due to lack of interpretability.

In this common case for science tasks, it is a necessity to use non-differentiable rewards. Our method does not preclude the use of differentiable rewards but rather offers a model-agnostic and plug-and-play inference-time strategy that is broadly applicable, even in low-data or black-box, or high-stakes applications. Furthermore, DSearch can technically integrate gradient information by using classifier guidance as proposal distribution (instead of pretrained models) to make the tree. Thus our method can be effectively integrated with gradient-based techniques.

## D Potential Limitations

Although totally controllable, our approach requires more computational resources (when not parallelized) or memory (when parallelized) than standard inference methods, as noted in Section 4. Taking this aspect into account, we compare DSearch, with baselines such as best-of-N in our experimental section (Section 6). For gradient-based approaches like classifier guidance and DPS, it is important to note that these methods also incur additional computational and memory complexity due to the backward pass, which DSearch avoids.

## E Variant: Dynamic Beam Resample

Under the strategy of dynamic search, we also explore an alternative design choice for beam control, as outlined in Algorithm 3. In this algorithm, we mitigate the waste of computational resources by replacing poor beams with high-quality ones, while both the beam width and the tree width can be fixed. Specifically, at each time step, after performing a greedy selection based on heuristic functions, we discard suboptimal beams of a certain percentage and resample from high-quality samples in Line 4 using the selection function, which samples by probability of exponential tiling. With beam replacement, DSearch-R drives extreme optimization at the expense of sample variability, while DSearch maintains a strong balance between diversity and reward optimization.

We compare the performance of DSearch and its variant DSearch-R with other methods. The main results are summarized in Table 2 on page 19. DSearch achieves superior reward performance across all evaluated tasks, consistently outperforming baselines. This trend is particularly evident in

---

**Algorithm 3** DSearch with Beam Resample (DSearch-R)

---

1: **Require**: Heuristic function $\{\hat{v}_t\}_{t=T}^0$, Search set $\mathcal{A}$, Beam width $b$, tree width $w$, resample rate $r_r$
2: **for** $t \in [T+1, \cdots, 1]$ **do**
3:    **if** $t \in \mathcal{A}$ **then**
4:      Do Line 4 in Algorithm 2.
5:      $v_{th} = Quantile_{1-r_r}(\{\hat{v}(z_{t-1}^{(j)})\}_{j=1}^b)$
6:      Drop beams and remain $B_r = \{z_{t-1}^{(j)} | \mathbb{1}(\hat{v}(z_{t-1}^{(j)}) \geq v_{th}) = 1\}$
7:      Resampling with replacement:

$$\{x_{t-1}^{(j)}\}_{j=1}^{r_r b} \sim \sum_{i=1}^{|B_r|} \frac{g(\hat{v}(z_{t-1}^{(i)}))}{\sum_{|B_r|} g(\hat{v}(z_{t-1}^{(i)}))} \delta(z_{t-1}^{(i)}),$$

     where $z_{t-1}^{(i)} \in B_r$, $g(\cdot) = \exp(\cdot/(\max_i \hat{v}(z_{t-1}^{(i)})))$.
8:      Remaining $\{x_{t-1}^{(j)}\}_{j=r_r b+1}^b = B_r$
9:    **else**
10:      $Ch(x_t^{(j)}) = x_{t-1}^{(j)} \sim p^{pre}(\cdot|x_t^{(j)})$
11:    **end if**
12: **end for**
13: **Output**: $\{x_0^{[j]}\}$

---

biological sequence generation tasks, where DSearch exhibits significantly higher scores in HepG2 enhancer activity, 5'UTR MRL, and stability. The improvement over methods such as Best-of-N, SVDD, and SMC suggests that DSearch's dynamic tree search effectively prioritizes high-reward samples while maintaining efficient exploration. DSearch-R, which employs beam replacement, exhibits an even stronger tendency to maximize rewards. However, as anticipated, this comes at the cost of reduced diversity, as the replacement mechanism strongly biases toward highly rewarding samples while discarding potential alternatives. While DSearch generally improves sample rewards, its naturalness remains competitive with baselines. In molecular generation tasks, DSearch achieves lower NLL compared to baselines, suggesting that it generates chemically realistic molecules. DSearch also exhibits a balance between diversity and reward, ensuring a reasonable level of diversity while significantly enhancing reward. In contrast, the baseline SMC and DSearch-R, which rely on batch resampling strategies, show a marked drop in diversity.

In Appendix G.1, we illustrate how DSearch and DSearch-R performances scale with computational budget $\bar{C}$.

## F  EXPERIMENTAL DETAILS

### F.1  TASK SETTINGS

#### F.1.1  IMAGES

We define compressibility score as the negative file size in kilobytes (kb) of the image after JPEG compression following (Black et al., 2023). We define aesthetic scorer implemented as a linear MLP on top of the CLIP embeddings, which is trained on more than 400k human evaluations. The human preference scorer Wu et al. (2023) is the CLIP model fine-tuned using an extensive dataset comprising 798,090 human ranking choices across 433,760 pairs of images. As pre-trained models, we use Stable Diffusion, which is a common text-to-image diffusion model. As prompts to condition, we use animal prompts following (Black et al., 2023) such as [Dog, Cat, Panda, Rabbit, Horse, ...] for aesthetic score task and human instruction prompts following (Wu et al., 2023) for HPS task.

#### F.1.2  MOLECULES

We calculate QED and SA scores using the RDKit (Landrum et al., 2016) library. We use the docking program QuickVina 2 (Alhossary et al., 2015) to compute the docking scores following Yang et al.

Table 2: Performance of different methods on alignment tasks w.r.t. reward, NLL/quality, and diversity. The computation budget $\bar{C}$ for the image compressibility, aesthetic and HPS tasks are 40, 45 and 55, Enhancer, 5'UTR MRL, and 5'UTR Stability tasks 100, 50, and 80, and molecular tasks 50, respectively. ↑ indicates higher values correspond to better performance while ↓ indicates lower for better. **bold** and underline highlight the best and second best performance, respectively.

| Method | Image Compressibility | | | Image Aesthetic Score | | | Image Human Preference Score | | |
| | Compressibility↑ | Quality↓ | Diversity↑ | Aesthetic↑ | Quality↓ | Diversity↑ | HPS↑ | Quality↓ | Diversity↑ |
|---|---|---|---|---|---|---|---|---|---|
| Pre-trained | -95.7 ± 7.8 | 11.4 ± 7.4 | 0.2852 ± 0.0301 | 5.45 ± 0.15 | 11.4 ± 7.4 | 0.2852 ± 0.0302 | 0.2729 ± 0.0037 | 14.5 ± 1.3 | 0.5161 ± 0.0476 |
| Best-N | -65.9 ± 3.4 | 24.0 ± 6.4 | 0.2972 ± 0.0283 | 6.25 ± 0.05 | 3.2 ± 2.3 | 0.2713 ± 0.0306 | 0.2907 ± 0.0006 | 12.1 ± 10.2 | 0.3182 ± 0.0322 |
| DPS | -61.0 ± 4.9 | 22.7 ± 1.3 | 0.2392 ± 0.0499 | 6.16 ± 0.07 | 6.1 ± 2.9 | 0.2875 ± 0.0184 | 0.2971 ± 0.0026 | 14.1 ± 3.5 | 0.4173 ± 0.0304 |
| SMC | -66.0 ± 7.8 | 21.9 ± 7.8 | 0.1825 ± 0.0791 | 6.08 ± 0.05 | 4.7 ± 0.8 | 0.0649 ± 0.0347 | 0.2771 ± 0.0015 | 17.6 ± 1.8 | 0.4445 ± 0.0230 |
| SVDD | -37.3 ± 6.6 | 46.7 ± 1.6 | 0.2758 ± 0.0363 | 6.37 ± 0.26 | 4.6 ± 5.7 | 0.2655 ± 0.0540 | 0.2970 ± 0.0051 | 22.1 ± 8.0 | 0.4577 ± 0.0144 |
| DSearch | -35.7 ± 4.2 | 42.7 ± 0.9 | 0.3156 ± 0.0111 | 6.54 ± 0.12 | 5.8 ± 10.5 | 0.2667 ± 0.0166 | **0.3133 ± 0.0058** | 16.5 ± 4.6 | 0.4323 ± 0.0534 |
| DSearch-R | **-21.6 ± 0.5** | 82.9 ± 3.1 | 0.1711 ± 0.0059 | **6.67 ± 0.08** | 1.8 ± 2.1 | 0.2020 ± 0.0041 | 0.2984 ± 0.0001 | 21.7 ± 2.0 | 0.3935 ± 0.0062 |

| Method | Enhancer HepG2 | | | 5'UTR MRL | | | 5'UTR Stability | | |
| | HepG2↑ | NLL↓ | Diversity↑ | MRL↑ | NLL↓ | Diversity↑ | Stability↑ | NLL↓ | Diversity↑ |
|---|---|---|---|---|---|---|---|---|---|
| Pre-trained | 0.305 ± 0.295 | 261.0 ± 0.5 | 0.7197 ± 0.1650 | 0.345 ± 0.112 | 68.4 ± 0.2 | 0.7380 ± 0.1263 | 0.212 ± 0.010 | 68.4 ± 0.3 | 0.7375 ± 0.1735 |
| Best-N | 3.319 ± 0.152 | 263.0 ± 0.8 | 0.7097 ± 0.1703 | 1.009 ± 0.006 | 68.0 ± 0.4 | 0.7280 ± 0.1248 | 0.342 ± 0.002 | 68.9 ± 0.2 | 0.7275 ± 0.1710 |
| DPS | 3.665 ± 0.222 | 258.0 ± 2.1 | 0.7454 ± 0.0755 | 0.995 ± 0.016 | 72.0 ± 0.2 | 0.7408 ± 0.0956 | 0.419 ± 0.002 | 67.0 ± 0.5 | 0.6040 ± 0.2188 |
| SMC | 5.601 ± 0.208 | 288.0 ± 1.0 | 0.5737 ± 0.3563 | 1.008 ± 0.013 | 68.5 ± 0.5 | 0.5544 ± 0.2857 | 0.329 ± 0.006 | 69.0 ± 0.6 | 0.4856 ± 0.4068 |
| SVDD | 7.040 ± 0.068 | 246.2 ± 5.3 | 0.7159 ± 0.1024 | 1.356 ± 0.009 | 66.7 ± 0.8 | 0.6349 ± 0.2027 | 0.469 ± 0.002 | 69.2 ± 0.8 | 0.7309 ± 0.1572 |
| DSearch | 7.245 ± 0.502 | 260.1 ± 1.9 | 0.7063 ± 0.1684 | 1.521 ± 0.011 | 68.6 ± 0.6 | 0.6258 ± 0.2135 | 0.533 ± 0.004 | 71.0 ± 0.7 | 0.7001 ± 0.1783 |
| DSearch-R | **8.149 ± 0.268** | 249.5 ± 3.8 | 0.6661 ± 0.3508 | **1.591 ± 0.006** | 66.9 ± 0.8 | 0.5236 ± 0.3051 | **0.573 ± 0.003** | 69.5 ± 0.8 | 0.5403 ± 0.3459 |

| Method | Molecule Drug-likeness | | | Molecule Synthetic Accessibility | | | Molecule Binding Affinity - Parp1 | | |
| | QED↑ | NLL↓ | Diversity↑ | SA↑ | NLL↓ | Diversity↑ | Docking Score↑ | NLL↓ | Diversity↑ |
|---|---|---|---|---|---|---|---|---|---|
| Pre-trained | 0.656 ± 0.007 | 958 ± 58 | 0.8733 ± 0.1580 | 0.652 ± 0.006 | 971 ± 69 | 0.8429 ± 0.2227 | 7.2 ± 0.5 | 971 ± 32 | 0.7784 ± 0.2998 |
| Best-N | 0.887 ± 0.008 | 943 ± 33 | 0.8779 ± 0.1579 | 0.921 ± 0.014 | 946 ± 62 | 0.8442 ± 0.2220 | 10.2 ± 0.4 | 951 ± 22 | 0.7938 ± 0.3052 |
| DPS | 0.885 ± 0.019 | 971 ± 41 | 0.8961 ± 0.0761 | 0.968 ± 0.026 | 917 ± 57 | 0.8968 ± 0.0752 | 11.6 ± 0.1 | 948 ± 63 | 0.8882 ± 0.0581 |
| SMC | 0.796 ± 0.007 | 1086 ± 21 | 0.6441 ± 0.2591 | 0.633 ± 0.007 | 1050 ± 28 | 0.6894 ± 0.2268 | 10.6 ± 0.5 | 957 ± 36 | 0.5092 ± 0.3673 |
| SVDD | 0.931 ± 0.003 | 1049 ± 24 | 0.8920 ± 0.0589 | 0.986 ± 0.019 | 1068 ± 24 | 0.8633 ± 0.2277 | 12.7 ± 0.2 | 993 ± 25 | 0.8980 ± 0.0635 |
| DSearch | **0.946 ± 0.002** | 911 ± 28 | 0.8424 ± 0.2195 | **1.000 ± 0.009** | 892 ± 61 | 0.8546 ± 0.2424 | 13.7 ± 0.3 | 731 ± 35 | 0.7650 ± 0.2934 |
| DSearch-R | 0.934 ± 0.001 | 527 ± 54 | 0.6145 ± 0.2053 | **1.000 ± 0.168** | 935 ± 31 | 0.4465 ± 0.3830 | **14.4 ± 0.2** | 647 ± 39 | 0.6871 ± 0.2555 |

(2021), with exhaustiveness as 1. Note that the docking scores are initially negative values, while we reverse it to be positive and then clip the values to be above 0, *i.e.*. We compute DS regarding protein parp1 (Poly [ADP-ribose] polymerase-1), which is a target protein that has the highest AUROC scores of protein-ligand binding affinities for DUD-E ligands approximated with AutoDock Vina.

### F.1.3 BIOLOGICAL SEQUENCES

We examine two publicly available large datasets: enhancers ($n \approx 700k$) (Gosai et al., 2023) and UTRs ($n \approx 300k$) (Sample et al., 2019), with activity levels measured by massively parallel reporter assays (MPRA) (Inoue et al., 2019), where the expression driven by each sequence is measured. These datasets have been widely used for sequence optimization in DNA and RNA engineering, particularly in advancing cell and RNA therapies (Castillo-Hair & Seelig, 2021; Lal et al., 2024; Ferreira DaSilva et al., 2024; Uehara et al., 2024b). We pretrain the masked discrete diffusion model (Sahoo et al., 2024) on all the sequences.

In the Enhancers dataset, each $x$ is a DNA sequence of length 200. The reward oracle is learned from this dataset using the Enformer architecture (Avsec et al., 2021), while $y \in \mathbb{R}$ is the measured activity in the HepG2 cell line. The Enformer trunk has 7 convolutional layers, each having 1536 channels. as well as 11 transformer layers, with 8 attention heads and a key length of 64. Dropout regularization is applied across the attention mechanism, with an attention dropout rate of 0.05, positional dropout of 0.01, and feedforward dropout of 0.4. The convolutional head for final prediction has 2*1536 input channels and uses average pooling, without an activation function. These datasets and reward models are widely used in the literature on computational enhancer design (Lal et al., 2024; Sarkar et al., 2024).

In the 5'UTRs dataset, $x$ is a 5'UTR RNA sequence of length 50. The reward oracles are learned from datasets using ConvGRU models (Dey & Salem, 2017), which has been widely acknowledged for computational RNA design, and $y \in \mathbb{R}$ is the mean ribosomal load (MRL) measured by polysome profiling, and the stability measured by half life (Agarwal & Kelley, 2022), respectively. The ConvGRU trunk has a stem input with 4 channels and a convolutional stem that outputs 64 channels using a kernel size of 15. The model contains 6 convolutional layers, each initialized with 64 channels and a kernel size of 5. The convolutional layers use ReLU as the activation function, and a residual connection is applied across layers. Batch normalization is applied to both the convolutional and GRU layers. A single GRU layer with dropout of 0.1 is added after the convolutional layers. The convolutional head for final prediction uses 64 input channels and average pooling, without batch normalization. Note that the stability reward is non-differentiable since the half life of 5'UTR is

measured after concatenation with coding regions and 3'Untranslated regions, following Agarwal & Kelley (2022).

## F.2 BASELINES DETAILS

We will explain in more detail how to implement baselines.

**SVDD.** For this baseline, we compare with SVDD-PM (Li et al., 2024). SVDD-PM directly use the reward feedback to evaluate, i.e., use $r(\hat{x}_0(x_t))$ as the estimated value function, which aligns with our usage for DSearch. The advantage of this approach is that no additional training is required as long as we have $r$. The duplication size is set for fair comparisons.

**DPS.** We require differentiable models. For this task, for those non-differentiable rewards in images, 5'UTRs and molecules, we need to learn differentiable estimations of the reward oracle using deep learning models. For images, we use standard CNNs for this purpose, which contain 3 residual blocks and use average pooling. For molecules, we follow the implementation in Lee et al. (2023), and we use Graph Isomorphism Network (GIN) model (Xu et al., 2018). In GIN, we use mean global pooling and the RELU activation function, and the dimension of the hidden layer is 300. The number of convolutional layers in the GIN model is selected from the set $\{3, 5\}$; and we select the maximum number of iterations from $\{300, 500, 1000\}$, the initial learning rate from $\{1e-3, 3e-3, 5e-3, 1e-4\}$, and the batch size from $\{32, 64, 128\}$. Note that we cannot compute derivatives with respect to adjacency matrices when using the GNN model. For the 5'UTR task, we use the ConvGRU model (Dey & Salem, 2017). The ConvGRU trunk has a stem input with 4 channels and a convolutional stem that outputs 64 channels using a kernel size of 15. The model contains 6 convolutional layers, each initialized with 64 channels and a kernel size of 5. The convolutional layers use ReLU as the activation function, and a residual connection is applied across layers. Batch normalization is applied to both the convolutional and GRU layers. A single GRU layer with dropout of 0.1 is added after the convolutional layers. The convolutional head for final prediction uses 64 input channels and average pooling, without batch normalization. For training, the batch size is selected from $\{16, 32, 64, 128\}$, the learning rate from $\{1e-4, 2e-4, 5e-4\}$, and the maximum number of iterations from $\{2k, 5k, 10k\}$. Regarding hyperparameter $\alpha$, we choose several candidates and report the best one. For image tasks we select from $\{5.0, 10.0\}$ and for bio-sequence tasks we select from $\{1.0, 2.0\}$. For molecule QED task we select from $\{0.2, 0.3, 0.4, 0.5\}$, for molecule SA task $\{0.1, 0.2, 0.3\}$, and for molecule docking tasks we select from $\{0.4, 0.5, 0.6\}$. The hyperparameters are chosen for good reward and diversity balance.

**SMC.** For value function models, we use the same method as SVDD-PM. Regarding $\alpha$, we choose several candidates and report the best one. For image tasks we select from $[10.0, 40.0]$. For Enhancer and 5'UTR tasks as well as molecule QED and SA tasks we select from $\{0.1, 0.2, 0.3, 0.4\}$, while for molecule docking tasks we select from $\{1.5, 2.0, 2.5\}$. The hyperparameters are chosen for good reward and diversity balance.

## F.3 METHOD IMPLEMENTATION DETAILS

We will explain in more detail how to implement our proposal. For DSearch, we control the search tree expansion with an initial width $w(T)$ and an initial over-sample rate $o(0) = N(0)/N(T)$, where $C = w(T) * N(T)$, and $N$ is the number of samples. Over the time steps, we use dynamic beam scheduling to gradually and strategically reduce $N(t)$, while maintain $C = w(t) * N(t)$, until we reach our defined final $N(0)$. The dynamic beam scheduling can be done using many algorithms; thus we regard it as a hyperparameter, which is detailed in the below subsections. In the main experiments, we select exponential beam scheduling. For DSearch-R, we control the search tree expansion with the computation budget $C = w * N$, which is of the same value as DSearch. At each time step, we use the selection function $g$ to resample and replace $rr * 100\%$ percent of suboptimal samples in the batch. We regard $rr$ as a hyperparameter which is selected from $\{0.03, 0.04, 0.05\}$. The dynamic search scheduling can be done using many functions; thus we regard it as a hyperparameter, which is detailed in the below subsections. In the main experiments, we select exponential search scheduling and control $|\mathcal{A}|/T = 65\%$. Note that to control the computational budget for fair comparisons with the baselines, we have not included look-ahead value estimation in the main results.

### F.3.1 DYNAMIC BEAM SCHEDULING

We use a progressive sample reduction strategy of dynamic beam scheduling to further optimize computational efficiency. This method dynamically reduces the number of candidate samples at each time step, starting with an over-sampled batch and gradually pruning less promising candidates. Such refinement aligns with the observation that early steps in diffusion are less critical, while later steps require greater precision (Li et al., 2024).

Let $N_0$ denote the initial sample size, $N_T$ the target batch size, and $t \in [1, T]$. At each step $t$, we maintain a sample size $N_t$ that decreases according to a predefined schedule, subject to $N_t \geq N_T$. We experiment with several reduction strategies:

- **Linear Reduction**: $N_t = \max\left(N_T, N_0 - t \cdot \frac{N_0 - N_T}{T}\right)$, where the sample size decreases linearly over time.

- **Exponential Decay**: $N_t = \max\left(N_T, N_0 \cdot \left(\frac{N_T}{N_0}\right)^{t/T}\right)$, ensuring faster reduction in early steps.

- **Quadratic Reduction**: $N_t = \max\left(N_T, N_T + (N_0 - N_T) \cdot \left(1 - \frac{t}{T}\right)^2\right)$, prioritizing sample diversity in early steps.

- **Sigmoid Reduction**: A smooth reduction, $N_t = \max\left(N_T, \frac{N_0}{1 + e^{-\kappa \cdot (t - T/2)}}\right)$, where $\kappa$ adjusts the steepness of the transition.

At each step, the scores of all candidates are evaluated using the reward estimation $\hat{r}(x)$. The top $N_t$ samples are retained for the next step, where:

$$\text{Selected Samples} = \underset{x_i \in \mathcal{X}}{\mathrm{argmax}} \{r(x_i)\}_{i=1}^{N_t}.$$

This approach ensures that computational resources are concentrated on high-quality candidates, aligning with the goals of diffusion decoding.

### F.3.2 SEARCH SCHEDULING

A key consideration in search-based inference for diffusion models is the efficient allocation of computational resources across diffusion time steps. Unlike the uniform search strategy employed in autoregressive search works, we incorporate a search scheduling mechanism that dynamically adjusts the computational effort during the diffusion process. This adjustment is motivated by the observation that early time steps often contain sparse information, while later time steps are more information-dense and critical for achieving accurate predictions.

We explore multiple scheduling strategies inspired by related work in reinforcement learning (Silver et al., 2016; Grill et al., 2020) and molecular design (Yang et al., 2020). Each strategy is parameterized to allow for flexibility, depending on the desired trade-off between computation and decoding quality.

Let $T$ represent the total number of time steps, $t \in [0, T - 1]$ the current time step, and $f(t)$ the frequency of search operations. The scheduling strategies are defined as follows:

- **Linear Scheduling**: Search frequency increases linearly with $t$, defined as $f(t) = \alpha \cdot t$, where $\alpha$ is a scaling factor.

- **Exponential Scheduling**: Search frequency grows exponentially, prioritizing later steps, given by $f(t) = e^{\beta \cdot t/T} - 1$, where $\beta$ controls the growth rate.

- **Step-Based Scheduling**: Searches are conducted at fixed intervals $I(t)$ that decrease over time. For example, search every $\lfloor T/(2^{t/T}) \rfloor$ steps.

- **Quadratic Scheduling**: A more gradual transition, given by $f(t) = \gamma \cdot (t/T)^2$, where $\gamma$ adjusts the quadratic scaling.

- **Sigmoid Scheduling**: A smooth transition, defined as $f(t) = \frac{1}{1 + e^{-\delta \cdot (t - T/2)}}$, where $\delta$ adjusts the steepness of the curve.

Each strategy dynamically modulates the computational intensity of search, with exact parameters $(\alpha, \beta, \gamma, \delta)$ chosen to control $|\mathcal{A}| = C^{\dagger}$ based on the computation budget. Empirical results demonstrate that some schedules significantly reduce computation while maintaining decoding quality.

The proposed search scheduling and progressive sample reduction strategies are integrated into diffusion models. By adaptively controlling the number of search operations and candidate samples, we achieve a balance between computational efficiency and decoding accuracy. Future work may explore adaptive learning methods to optimize these schedules dynamically.

### F.4    SOFTWARE AND HARDWARE

Our implementation is under the architecture of PyTorch (Paszke et al., 2019). The deployment environments are Ubuntu 20.04 with 48 Intel(R) Xeon(R) Silver, 4214R CPU @ 2.40GHz, 755GB RAM, and graphics cards NVIDIA RTX 2080Ti. Each of our image experiments is conducted on a single A100 GPU, while Each of our experiments on other tasks on a single NVIDIA RTX 2080Ti or RTX A6000 GPU.

### F.5    LICENSES

The pretrained models and datasets for image tasks are under MIT license and Apache license 2.0, respectively. The dataset for molecular tasks is under Database Contents License (DbCL) v1.0. The dataset for DNA task is covered under AGPL-3.0 license. The dataset for RNA tasks is under GPL-3.0 license. We follow the regulations for all licenses.

# G    EXPERIMENTAL ANALYSIS AND STUDIES

## G.1    PERFORMANCE SCALING FOR ALL TASKS

Figure 7 and Figure 8 illustrates how DSearch performance scales with computational budge across all tasks. Similarly, As $\bar{C}$ increases, reward scores improve for all methods, but the gains are most pronounced for DSearch and DSearch-R. This shows that dynamic tree search effectively utilizes additional computation to align samples.

## G.2    VISUALIZATION OF TREE WIDTH AND BEAM WIDTH IN DSEARCH

To better understand the impact of DSearch as well as our proposed exponential search scheduling and beam scheduling in an actual task setting, we visualize the evolution of tree width $w(t)$ and beam width $b(t)$ during the molecule optimization process under a controlled computational budget of $\bar{C} = 55$. As can be observed in Figure 9, the right side shows the corresponding beam width $b(t)$, representing the number of retained candidates at each step, while the left side of the figure illustrates the variation of tree width $w(t)$, which determines the number of candidate expansions at each step. As time progresses, the beam width is progressively reduced using exponential beam scheduling, ensuring computational resources are concentrated on high-quality candidates. Meanwhile, the search tree width dynamically expands following an exponential growth strategy, prioritizing later steps where higher reward regions are more effectively explored. We can also observe the exponential search scheduling, where tree width is 1 at some time steps, particularly earlier ones. The figure implies how these scheduling strategies practically balance exploration and exploitation, improving search efficiency while maintaining diversity in generated molecules.

## G.3    RUNTIME AND COMPUTATIONAL COMPLEXITY STUDIES

In Section 4, we have analyzed the computational complexity of DSearch, which is $O(T\bar{C})$, where $\bar{C} = (|\mathcal{A}|C + T - |\mathcal{A}|)/T$, considering the time complexity of one diffusion inference time step as the unit. While we have theoretically ensured that DSearch and baseline methods operate under the same computational budget $\bar{C}$, practical execution time may vary due to implementation details, memory efficiency, and computational overhead. To empirically validate the runtime efficiency of DSearch, we compare its execution time against baseline SVDD across different values of $\bar{C}$, which represents the computational budget allocated for inference. The results are shown in Table

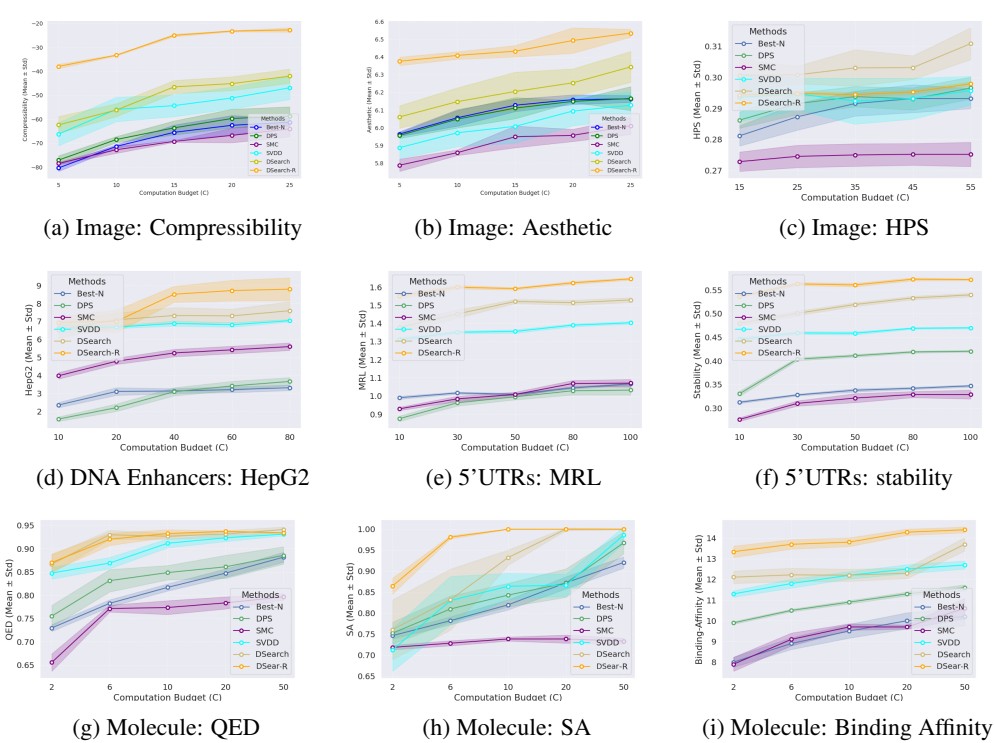

Figure 7: Reward (median & standard deviation) under different constraints $\bar{C}$.

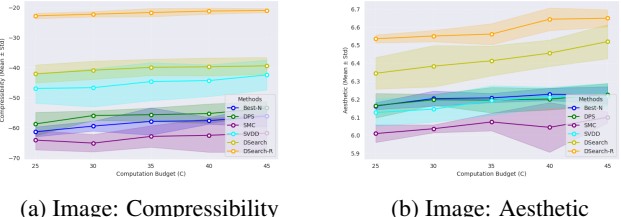

Figure 8: Reward (median & standard deviation) under larger constraints $\bar{C}$ (25-45).

3 on page 24. At lower computation budgets ($\bar{C}$=10), DSearch incurs a slightly higher execution time than SVDD. This overhead is expected, as DSearch dynamically adjusts beam search width, introducing additional computations beyond simple intermediate state selection like SVDD. As $\bar{C}$ increases to 20, the execution times of both methods become more comparable. This suggests that the initial overhead of DSearch becomes less significant relative to the overall computation. At higher computation budgets ($\bar{C} \geq 40$), DSearch achieves reduction in execution time compared to SVDD. This demonstrates that DSearch scales more efficiently as computation increases, likely due to its adaptive beam scheduling, which reduces the total number of samples. This empirical study reinforces the theoretical claims that DSearch not only matches but might surpasses the efficiency of other alignment methods making it a compelling choice for structured sequence and molecule generation tasks.

We also provide plot with time as x-axis beyond time comparisons. To enable this, we run under multiple budgets for all methods to choose similar runtimes, while data points are unevenly distributed due to the uncontrollableness of precise runtime Figure 10 shows DSearch and DSearch-R scale significantly faster in reward per second, much more efficient than DPS and Best-of-N. These results further confirm that DSearch retains better cost-effectiveness.

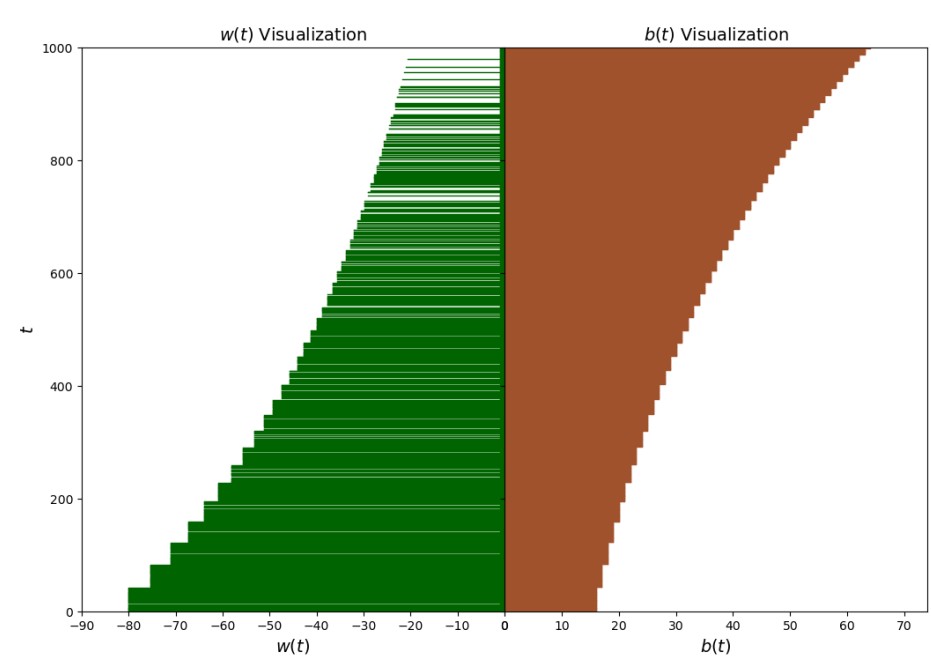

Figure 9: Visualization of dynamic tree width and beam width change in molecular task.

Table 3: Runtime comparison (in seconds) across different methods and computational budgets $\bar{C}$. All methods are evaluated under consistent hardware. Runtimes are from measured wall-clock time. For DPS and SMC, we apply Best-of-N (N=$\bar{C}$) on top for fair comparison regarding computations.

| Method | $\bar{C} = 10$ | $\bar{C} = 20$ | $\bar{C} = 40$ | $\bar{C} = 60$ | $\bar{C} = 80$ |
|---|---|---|---|---|---|
| Best-of-N | 27.56 | 63.24 | 94.81 | 135.46 | 183.55 |
| DPS | 81.14 | 155.40 | 309.62 | 460.27 | 564.05 |
| SMC | 55.80 | 111.21 | 224.46 | 326.44 | 432.81 |
| SVDD | **23.36** | **44.93** | 88.52 | 128.14 | 172.41 |
| DSearch | 31.43 | 48.31 | 81.78 | 116.20 | 145.87 |
| DSearch-R | 27.78 | 51.12 | **77.90** | **104.70** | **140.22** |

### G.4 REWARD ESTIMATION ANALYSIS

To show the quality of our estimated value functions, i.e., heuristic functions, we evaluate the effectiveness of our value estimation method for predicting the final reward of diffusion-generated samples at intermediate time steps. Since the true reward is only available at the final state $x_0$, we assess the accuracy of our intermediate state value predictions by computing the Pearson correlation coefficient between the estimated reward and the actual reward obtained at $x_0$. We visualize this relationship using scatter density plots across several time steps, illustrating how well the estimated reward aligns with the expected ground-truth reward. For each sampled trajectory, we estimate rewards at various intermediate diffusion steps and compare them against the final ground-truth reward. Specifically, we track the correlation at time steps 32, 64, 88, 112, 116, 120, 124, and 127, covering a range from early diffusion stages to the final steps. The Pearson correlation coefficient is used as a measure of how well the estimated rewards predict the final reward. Higher values of Pearson correlation indicate better alignment between the estimated and actual rewards.

As shown in Figures 11, 12, 13, during early diffusion steps, the estimated rewards show a weak correlation with the final reward, suggesting that at early stages they carry limited predictive power. This is expected, as diffusion-based generation starts from a highly noisy prior, and meaningful structure has not yet emerged. In mid diffusion steps the correlation improves noticeably, indicating that as the denoising process progresses, the estimated reward begins to capture useful information about the final state. The scatter plots show that the spread of points starts to concentrate along the

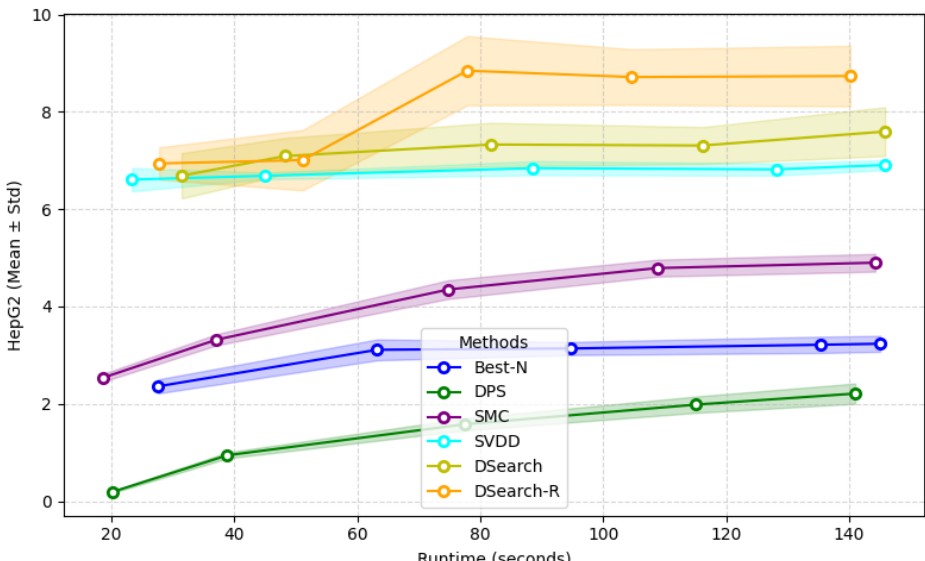

Figure 10: Reward (median & standard deviation) comparisons across different methods under different runtimes. We run all experiments on the same hardware to obtain comparisons within a certain time region, though the data points are unevenly distributed due to the uncontrollableness of the precise runtime.

diagonal, reflecting a stronger relationship between estimated and actual rewards. At late diffusion steps, the estimated rewards achieve a high correlation with the final reward. At $T = 127$, the correlation is nearly perfect, confirming that by the end of the diffusion process, our value estimation method accurately predicts the final reward. The density of points along the red diagonal line suggests that the estimated values are well-calibrated. The strong correlation in later steps supports the effectiveness of using this value function for intermediate state selection in our search strategies.

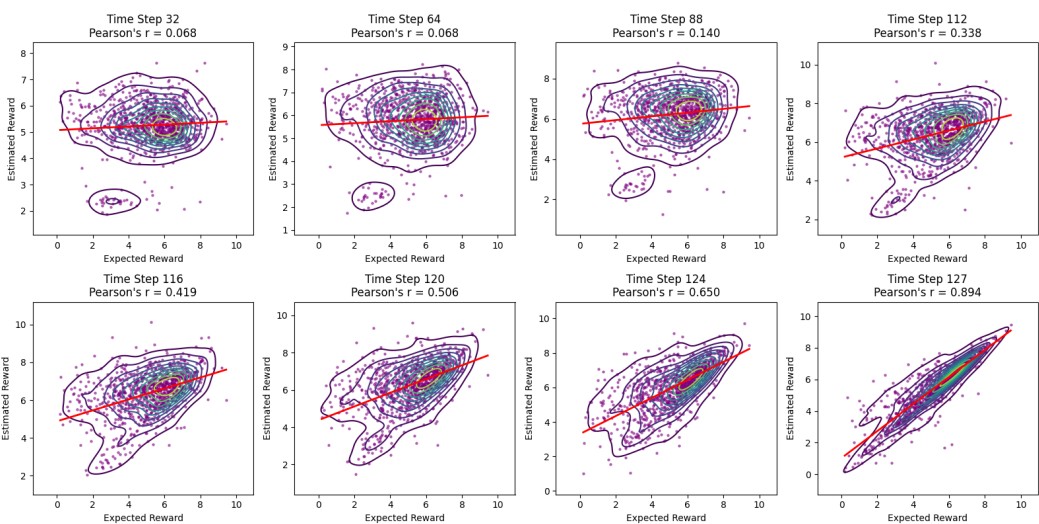

Figure 11: Scatter density plots between estimated reward and ground truth reward for DNA Enhancer task.

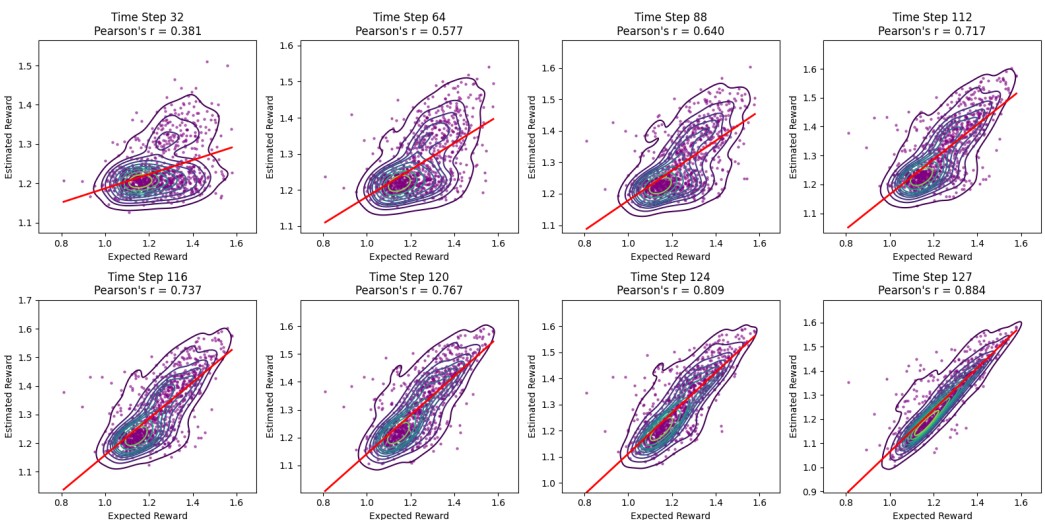

Figure 12: Scatter density plots between estimated reward and ground truth reward for 5'UTR MRL task.

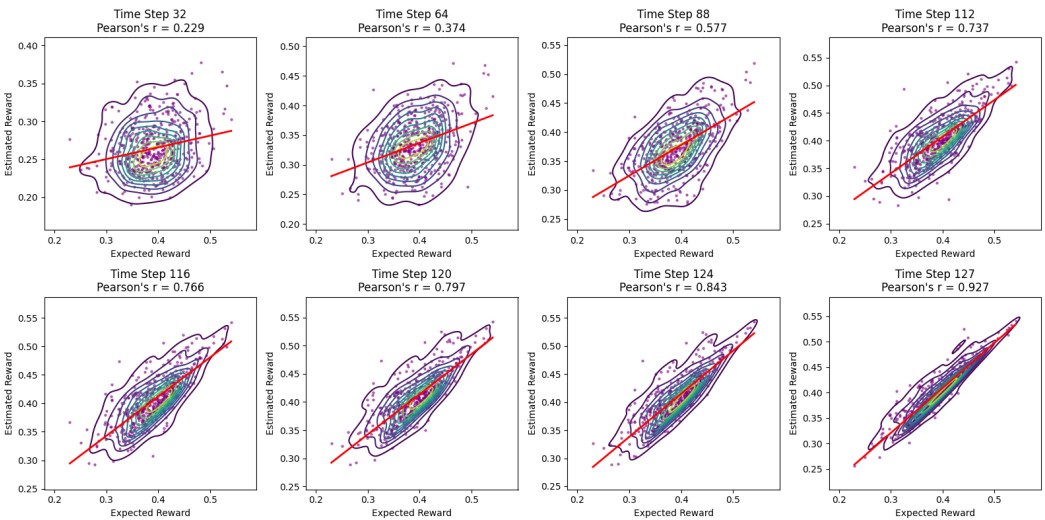

Figure 13: Scatter density plots between estimated reward and ground truth reward for 5'UTR stability task.

### G.5  MORE METRICS FOR MOLECULE GENERATION.

To further evaluate the validity of our method in molecule generation, we report several key metrics that capture different aspects of molecule quality and diversity in Table 4 on page 26.

Table 4: Comparison of the generated molecules across various metrics. The best values for each metric are highlighted in **bold**.

| Method | Valid↑ | Unique↑ | Novelty↑ | FCD↓ | SNN↑ | Frag↑ | Scaf↑ | NSPDK MMD↓ | Mol Stable↑ | Atm Stable↑ |
|---|---|---|---|---|---|---|---|---|---|---|
| Pre-trained | **1.000** | **1.000** | **1.000** | 12.979 | **0.414** | 0.513 | **1.000** | **0.038** | 0.320 | **0.917** |
| DPS | **1.000** | **1.000** | **1.000** | 13.230 | 0.389 | 0.388 | **1.000** | 0.040 | 0.310 | 0.878 |
| SMC | **1.000** | 0.406 | **1.000** | 22.710 | 0.225 | 0.068 | **1.000** | 0.285 | 0.000 | **0.968** |
| SVDD | **1.000** | **1.000** | **1.000** | 14.765 | 0.349 | 0.478 | **1.000** | 0.063 | **0.375** | 0.932 |
| DSearch | **1.000** | **1.000** | **1.000** | 13.305 | 0.389 | 0.412 | **1.000** | 0.086 | 0.200 | 0.902 |
| DSearch-R | **1.000** | 0.766 | **1.000** | **11.873** | 0.344 | **0.519** | **1.000** | 0.117 | 0.030 | 0.891 |

The validity of a molecule indicates its adherence to chemical rules, defined by whether it can be successfully converted to SMILES strings by RDKit. Uniqueness refers to the proportion of generated molecules that are distinct by SMILES string. Novelty measures the percentage of the generated molecules that are not present in the training set. Fréchet ChemNet Distance (FCD) measures the similarity between the generated molecules and the test set. The Similarity to Nearest Neighbors (SNN) metric evaluates how similar the generated molecules are to their nearest neighbors in the test set. Fragment similarity measures the similarity of molecular fragments between generated molecules and the test set. Scaffold similarity assesses the resemblance of the molecular scaffolds in the generated set to those in the test set. The neighborhood subgraph pairwise distance kernel Maximum Mean Discrepancy (NSPDK MMD) quantifies the difference in the distribution of graph substructures between generated molecules and the test set considering node and edge features. Atom stability measures the percentage of atoms with correct bond valencies. Molecule stability measures the fraction of generated molecules that are chemically stable, *i.e.*, whose all atoms have correct bond valencies. Specifically, atom and molecule stability are calculated using conformers generated by RDKit and optimized with UFF (Universal Force Field) and MMFF (Merck Molecular Force Field).

We compare the metrics using 512 molecules generated from the pre-trained GDSS model and from different methods optimizing QED, as shown in Table 4 on page 26. Overall, DSearch achieves comparable performances with the pre-trained model and other baselines, maintaining high validity, novelty, and uniqueness while outperforming on several metrics such as FCD and fragment similarity. DSearch-R achieves the best FCD (distribution similarity) but sacrifices stability. SVDD achieves a good balance between FCD, fragment similarity, and stability. SMC performs poorly in fragment similarity, NSPDK MMD, and molecular stability, indicating that it generates unrealistic molecules. Pre-trained performs consistently well across all metrics, particularly in SNN and atomic stability. However, it does not optimize specific molecular properties as effectively as the other methods. These results indicate that our approach can generally generate a diverse set of novel molecules that are chemically plausible and relevant.

## H    FURTHER EXPERIMENTAL RESULTS

### H.1    REWARD HISTOGRAMS

In the main text, we present the medians. Here, we plot the reward score distributions of generated samples as histograms, shown in Figure 14 - Figure 22.

### H.2    MORE ABLATION STUDIES ON THE EFFECTIVENESS OF SCHEDULING

To improve the efficiency of the search process, we apply scheduling search expansion. In diffusion-based sampling, earlier time steps contribute less to the final quality of the generated sequences, while later time steps contain more crucial information. To exploit this property, Search Scheduling dynamically adjusts the frequency of search operations, allocating more resources where they are most impactful. For search scheduling, we compare different scheduling strategies, including uniform, linear, exponential, and no search schedules, and evaluate how well they balance computational efficiency and performance. As shown in Figure 23, we observe that exponential scheduling achieves better rewards while reducing computational cost by 35% compared to the no scheduling ("all") baseline. This suggests that focusing search efforts in the later steps of the generation process leads to better sample quality without requiring a proportional increase in computation. Linear and uniform scheduling also improve efficiency but do not reach the same level of performance, as they distribute search operations more evenly across time steps and inefficiently expends resources. These results validate that adaptive scheduling allows for significant computational savings while maintaining or even improving generation quality. The effectiveness of exponential scheduling suggests that prioritizing late-stage refinement leads to better sample optimization, highlighting the importance of strategic search allocation in diffusion-based methods. Beam scheduling aims to improve sample selection by initially generating a larger batch of candidates and then progressively pruning weaker samples at intermediate steps. Instead of treating all samples equally throughout the entire diffusion process, this approach selectively retains high-quality candidates, allowing computational resources to be focused on the most promising sequences. We evaluate different beam scheduling strategies, including quadratic, linear, sigmoid, exponential, and no pruning schedules. From Figure 23, we observe that dropping weaker samples through exponential beam scheduling performs the best. This

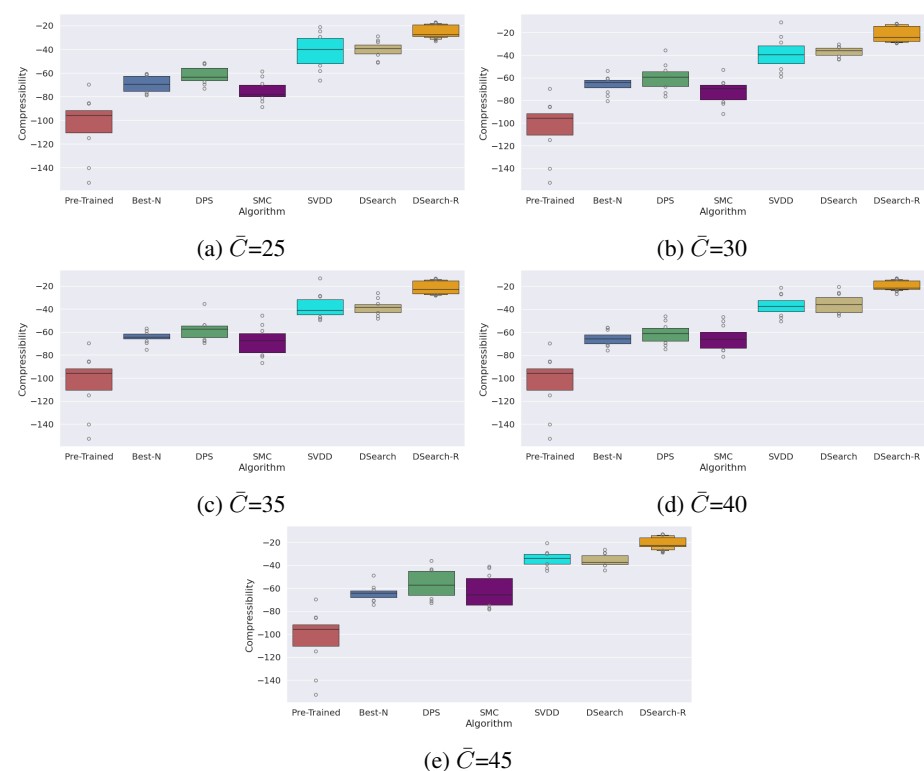

(a) $\bar{C}$=25

(b) $\bar{C}$=30

(c) $\bar{C}$=35

(d) $\bar{C}$=40

(e) $\bar{C}$=45

Figure 14: We show the histogram of generated samples in terms of rewards in compressibility of images. We consistently observe that our method demonstrates strong performances.

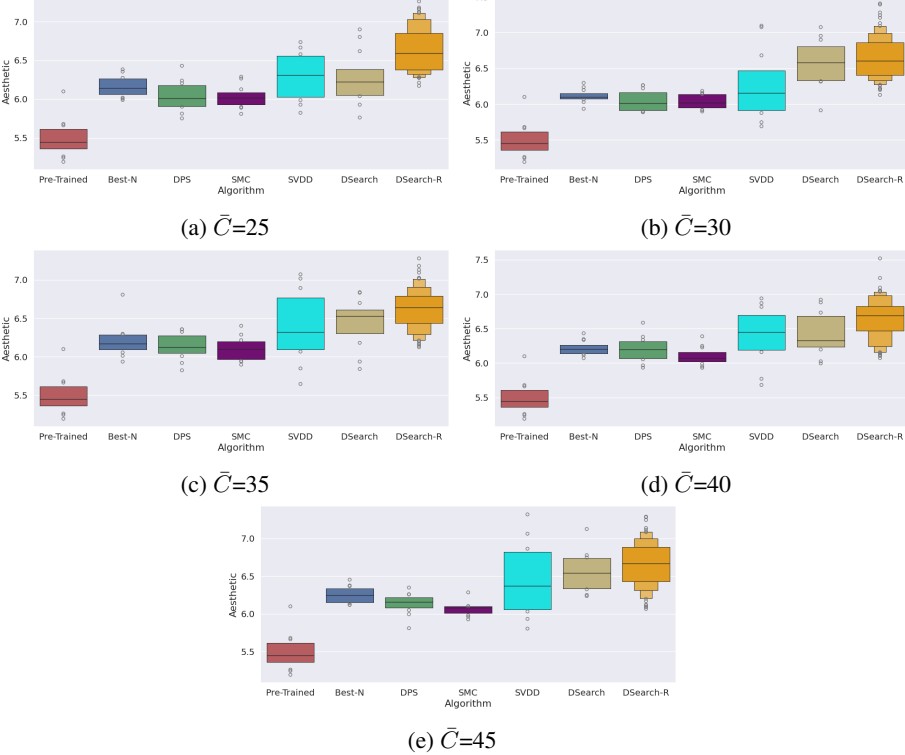

(a) $\bar{C}$=25

(b) $\bar{C}$=30

(c) $\bar{C}$=35

(d) $\bar{C}$=40

(e) $\bar{C}$=45

Figure 15: We show the histogram of generated samples in terms of rewards in aesthetic score of images. We consistently observe that our method demonstrates strong performances.

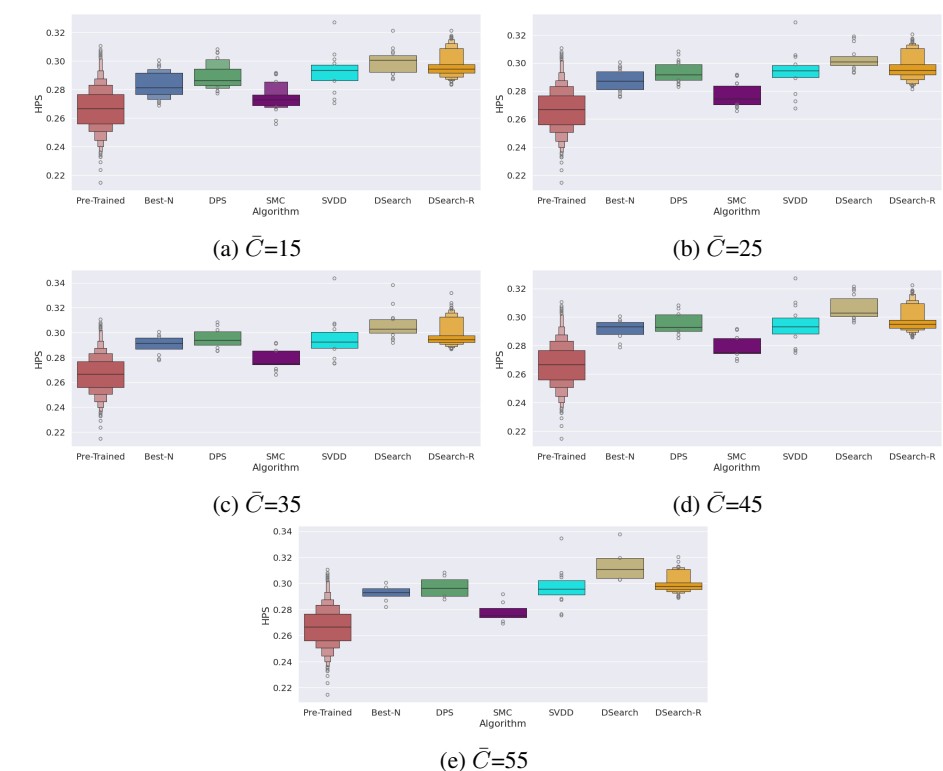

(a) $\bar{C}$=15

(b) $\bar{C}$=25

(c) $\bar{C}$=35

(d) $\bar{C}$=45

(e) $\bar{C}$=55

Figure 16: We show the histogram of generated samples in terms of rewards in human preference score of images. We consistently observe that our method demonstrates strong performances.

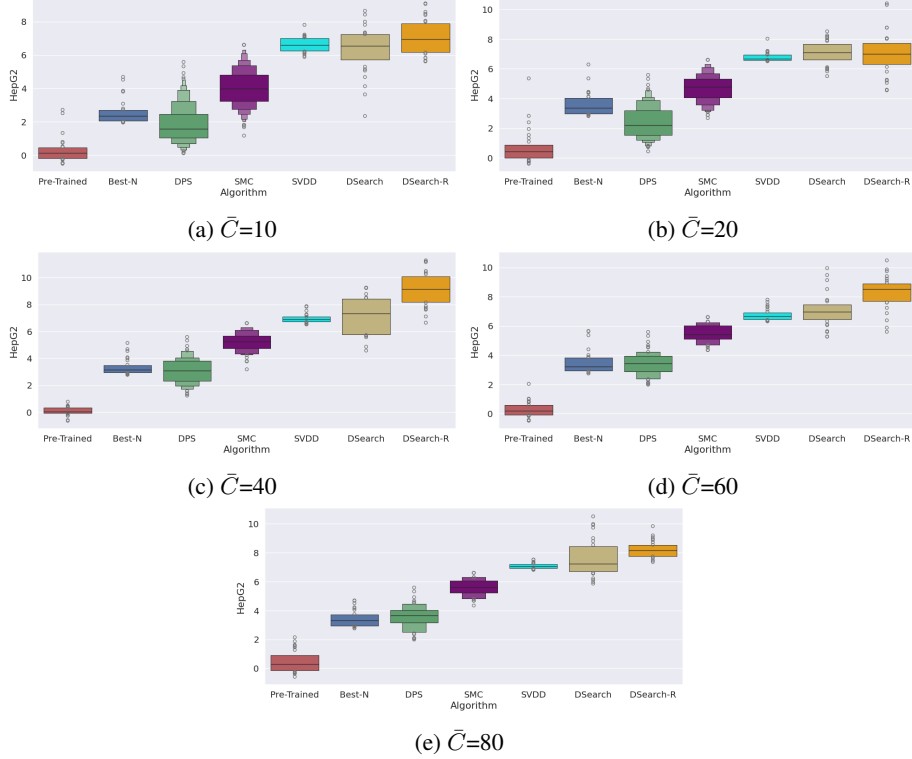

(a) $\bar{C}$=10

(b) $\bar{C}$=20

(c) $\bar{C}$=40

(d) $\bar{C}$=60

(e) $\bar{C}$=80

Figure 17: We show the histogram of generated samples in terms of rewards in HepG2 of DNA Enhancers. We consistently observe that our method demonstrates strong performances.

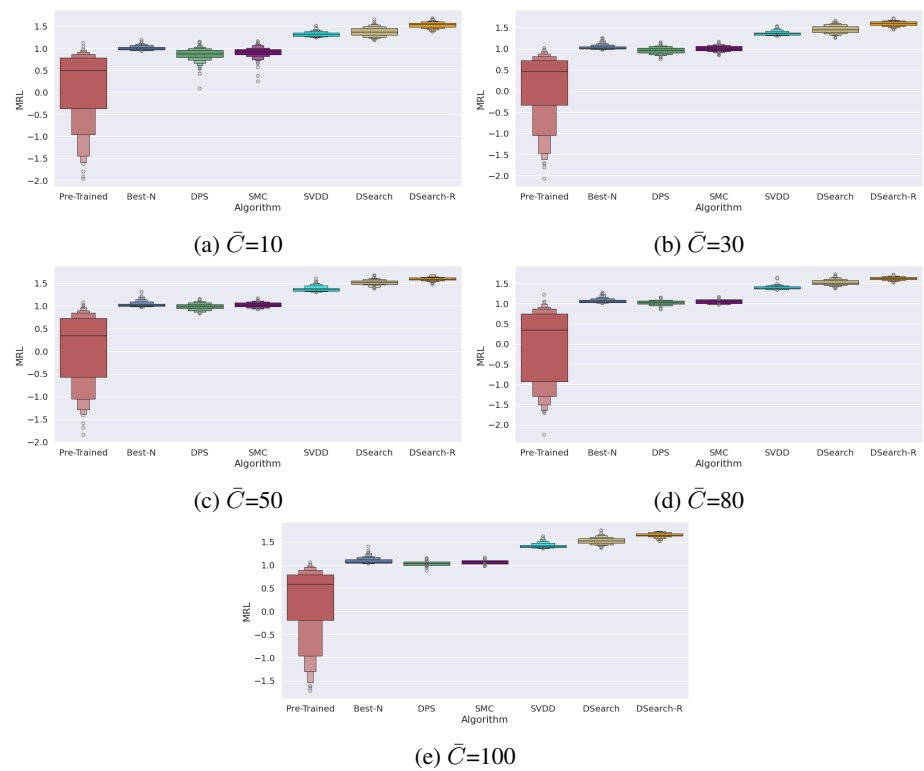

Figure 18: We show the histogram of generated samples in terms of rewards in MRL of 5'UTRs. We consistently observe that our method demonstrates strong performances.

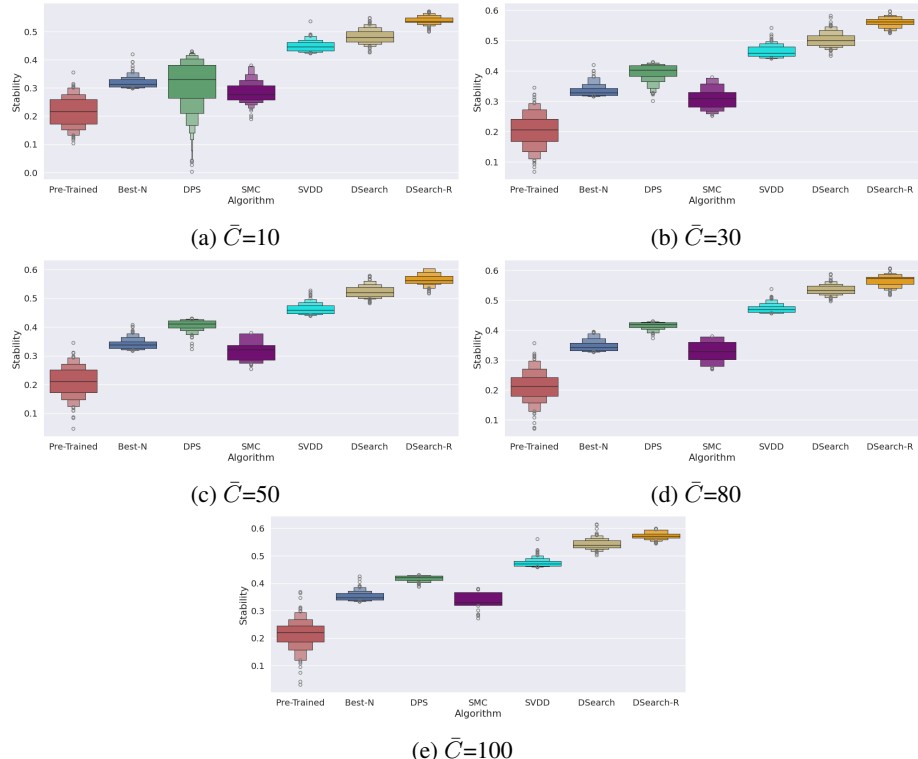

Figure 19: We show the histogram of generated samples in terms of rewards in stability of 5'UTRs. We consistently observe that our method demonstrates strong performances.

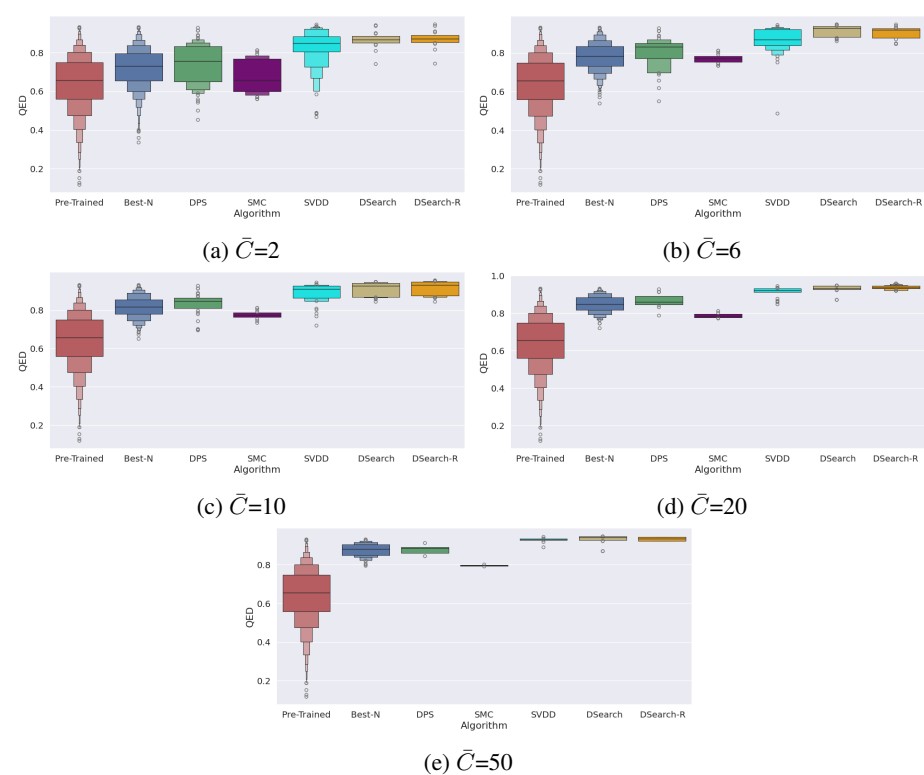

Figure 20: We show the histogram of generated samples in terms of rewards in QED of molecules. We consistently observe that our method demonstrates strong performances.

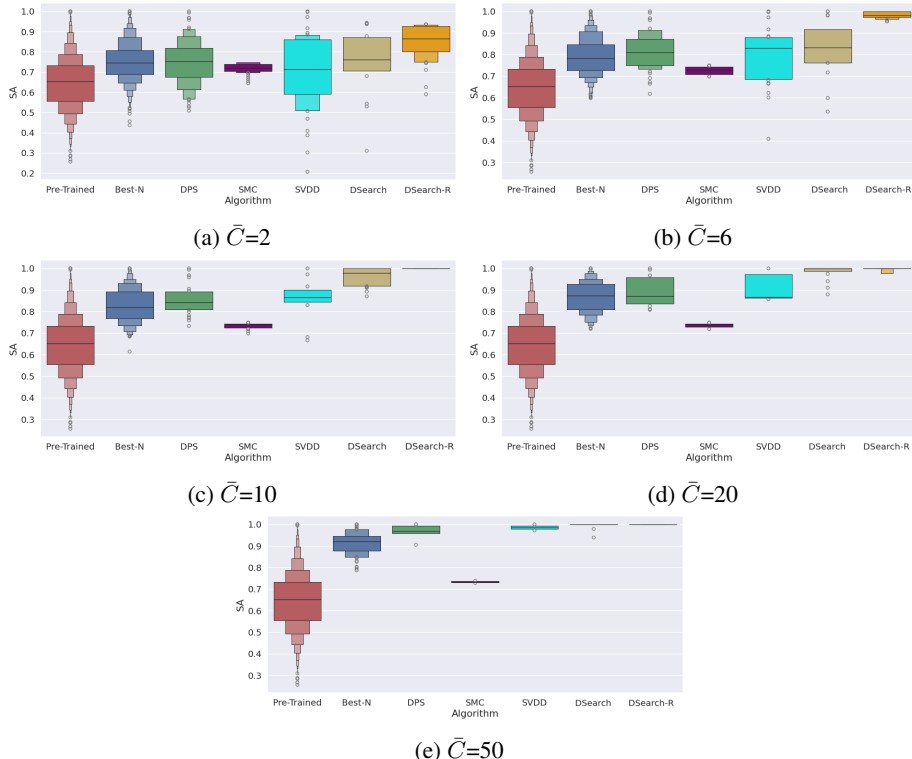

Figure 21: We show the histogram of generated samples in terms of rewards in SA of molecules. We consistently observe that our method demonstrates strong performances.

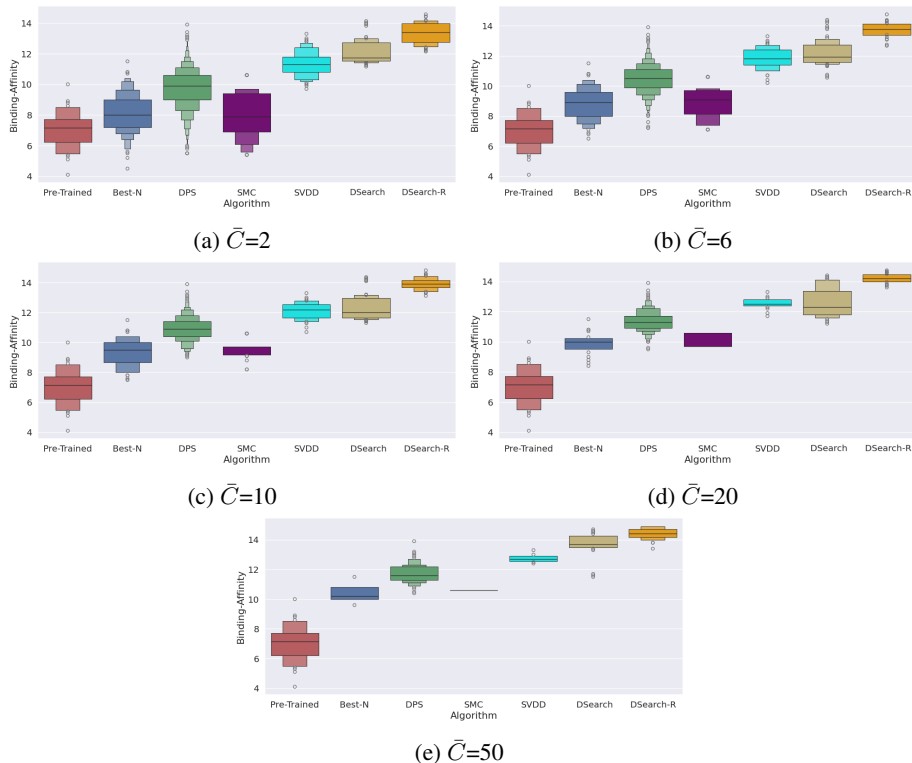

Figure 22: We show the histogram of generated samples in terms of rewards in binding affinity of molecules. We consistently observe that our method demonstrates strong performances.

demonstrating that reducing the search space aggressively in earlier steps allows for wider and more refined exploration later in the process, adapting to the dynamic nature of the search. In contrast, linear pruning strategies lead to suboptimal results, likely because they remove candidates at a fixed rate rather than adapting to the dynamic nature of the search. These results indicate that progressively focusing efforts on high-quality samples enhances overall alignment performance without increasing computational overhead, and dynamic beam reduction is a key factor in DSearch. Exponential beam pruning is particularly effective, as it ensures that early-stage candidates are explored broadly while later-stage refinement is performed on only the most promising samples. This confirms that dynamic beam reduction is a key factor in improving sample quality without increasing computational overhead.

### H.3 MORE ABLATION STUDIES ON THE EFFECTIVENESS OF LOOK AHEAD VALUE ESTIMATION

Lookahead mechanism strengthens the reward estimation of intermediate states. We explore the impact of different lookahead horizons $K$. For each sample, we generate $M = 6$ lookaheads of $K$ steps, compute the corresponding final rewards, and select the best intermediate states either by the maximum of these evaluations. From Figure 24, we observe that increasing $K$ consistently improves performance across different tasks, as it allows for a more informed selection of intermediate states. However, the gains saturate beyond a certain threshold, suggesting a limit of gain from the estimation accuracy.

### H.4 VISUALIZATION OF GENERATED SAMPLES

We provide additional generated samples in this section. Figure 25, Figure 26, and Figure 27 show comparisons of generated images from baseline methods and DSearch regarding compressibility, aesthetic score, and HPS, respectively. Figure 28 and Figure 29 presents the comparisons of visualized molecules generated from the baseline methods and DSearch regarding QED and SA, respectively. The visualizations validate the strong performances of DSearch, showing that DSearch can achieve

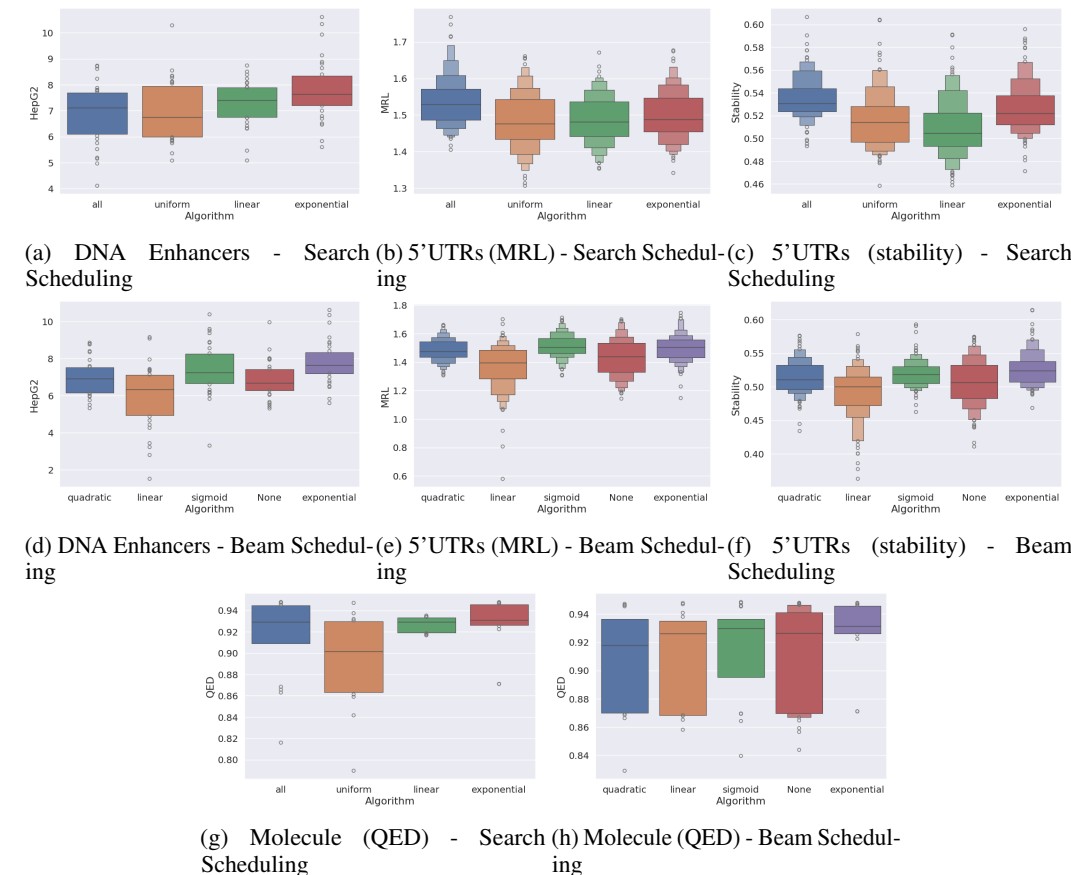

(a) DNA Enhancers - Search Scheduling

(b) 5'UTRs (MRL) - Search Scheduling

(c) 5'UTRs (stability) - Search Scheduling

(d) DNA Enhancers - Beam Scheduling

(e) 5'UTRs (MRL) - Beam Scheduling

(f) 5'UTRs (stability) - Beam Scheduling

(g) Molecule (QED) - Search Scheduling

(h) Molecule (QED) - Beam Scheduling

Figure 23: We show the reward distributions of generated samples using DSearch with different scheduling hyper-selections.

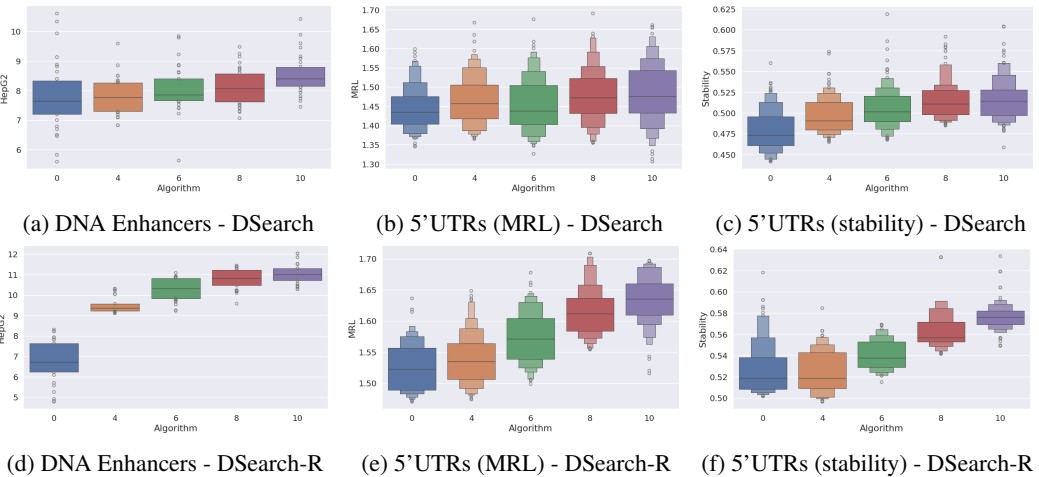

(a) DNA Enhancers - DSearch

(b) 5'UTRs (MRL) - DSearch

(c) 5'UTRs (stability) - DSearch

(d) DNA Enhancers - DSearch-R

(e) 5'UTRs (MRL) - DSearch-R

(f) 5'UTRs (stability) - DSearch-R

Figure 24: We show the reward distributions of generated samples with different $K$ values.

optimal SA for many molecules. In Figure 30, and Figure 31 we visualizes the docking of DSearch-generated molecular ligands to protein parp1. Docking scores presented above each column quantify the binding affinity of the ligand-protein interaction, while the figures include various representations and perspectives of the ligand-protein complexes. We aim to provide a complete picture of how each ligand is situated within both the local binding environment and the larger structural framework of the protein. First rows show close-up views of the ligand bound to the protein surface, displaying

the topography and electrostatic properties of the protein's binding pocket and providing insight into the complementarity between the ligand and the pocket's surface. Second rows display distant views of the protein using the surface representation, offering a broader perspective on the ligand's spatial orientation within the global protein structure. Third rows provide close-up views of the ligand interaction using a ribbon diagram, which represents the protein's secondary structure, such as alpha-helices and beta-sheets, to highlight the specific regions of the protein involved in binding. Fourth rows show distant views of the entire protein structure in ribbon diagram, with ligands displayed within the context of the protein's full tertiary structure. Ligands generally fit snugly within the protein pocket, as evidenced by the close-up views in both the surface and ribbon diagrams, which show minimal steric clashes and strong surface complementarity.

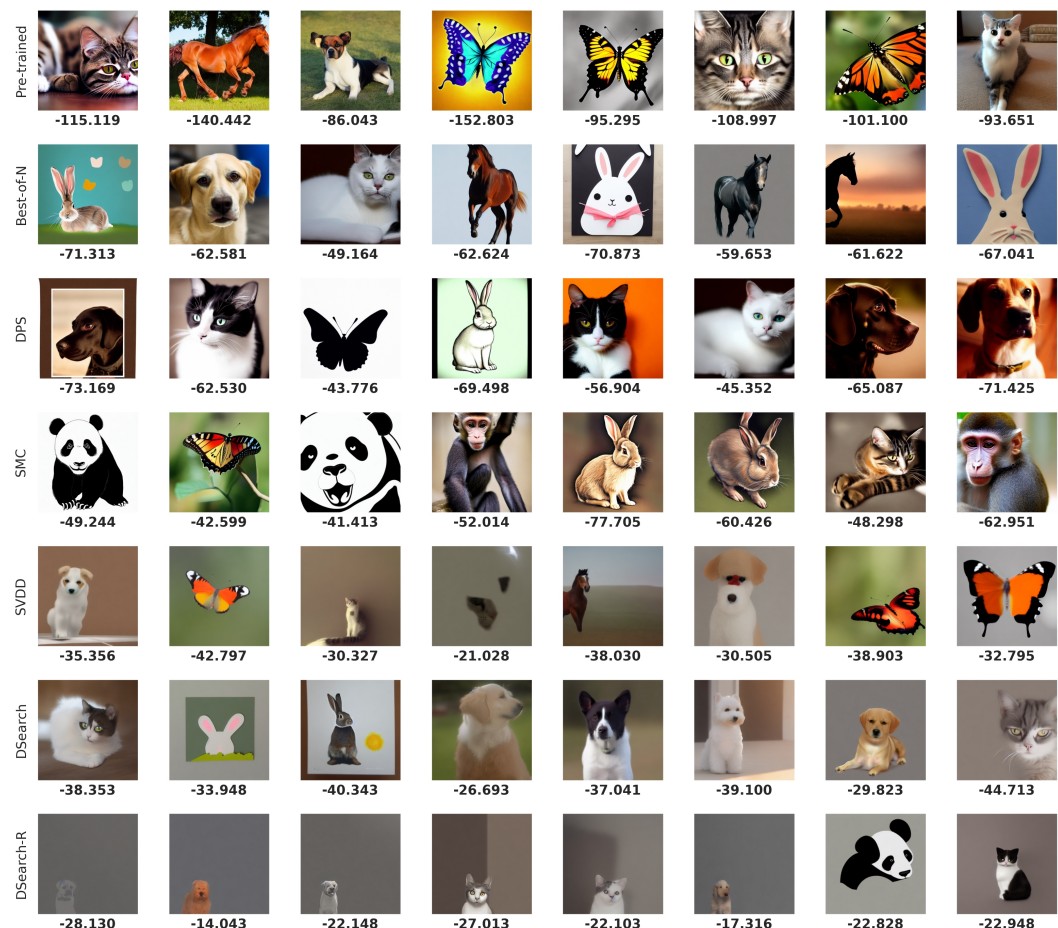

Figure 25: Visualization of generated images using different methods optimizing the reward of compressibility.

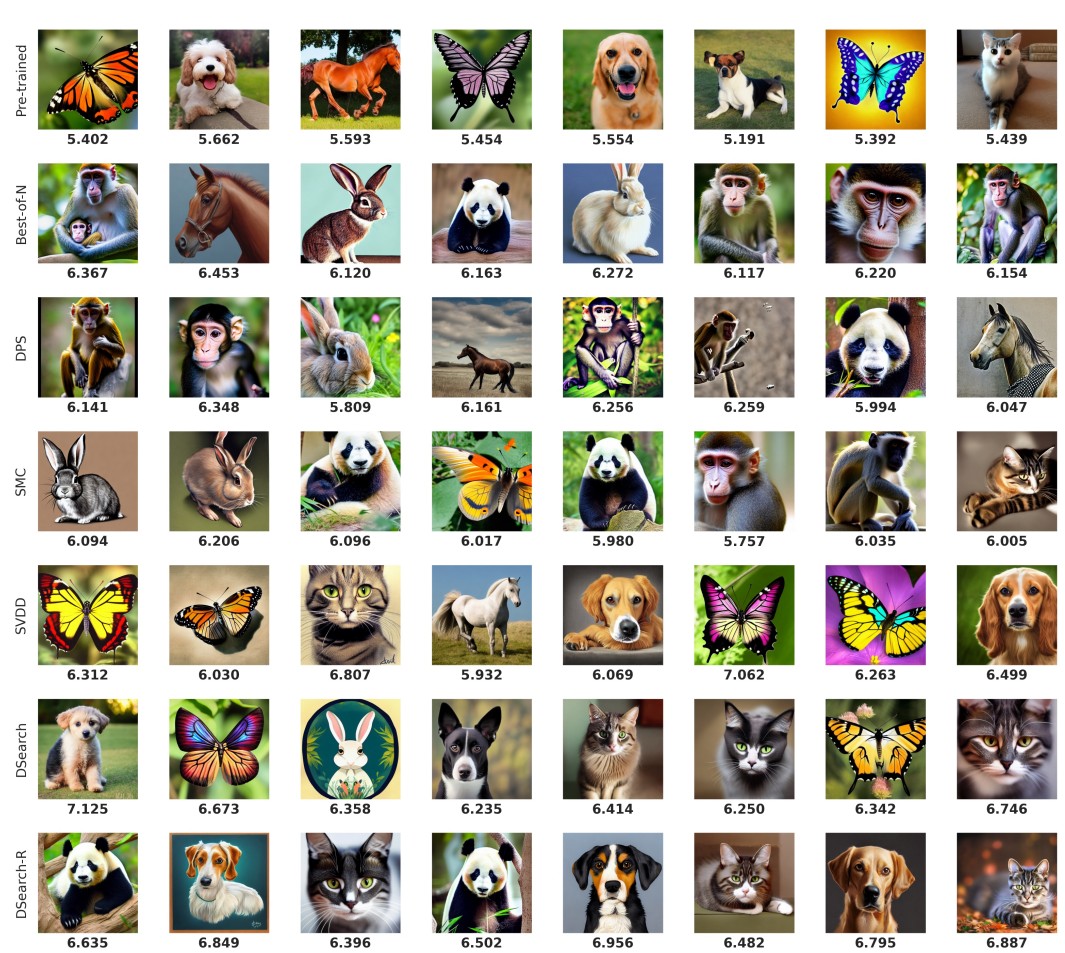

Figure 26: Visualization of generated images using different methods optimizing the reward of aesthetic score.

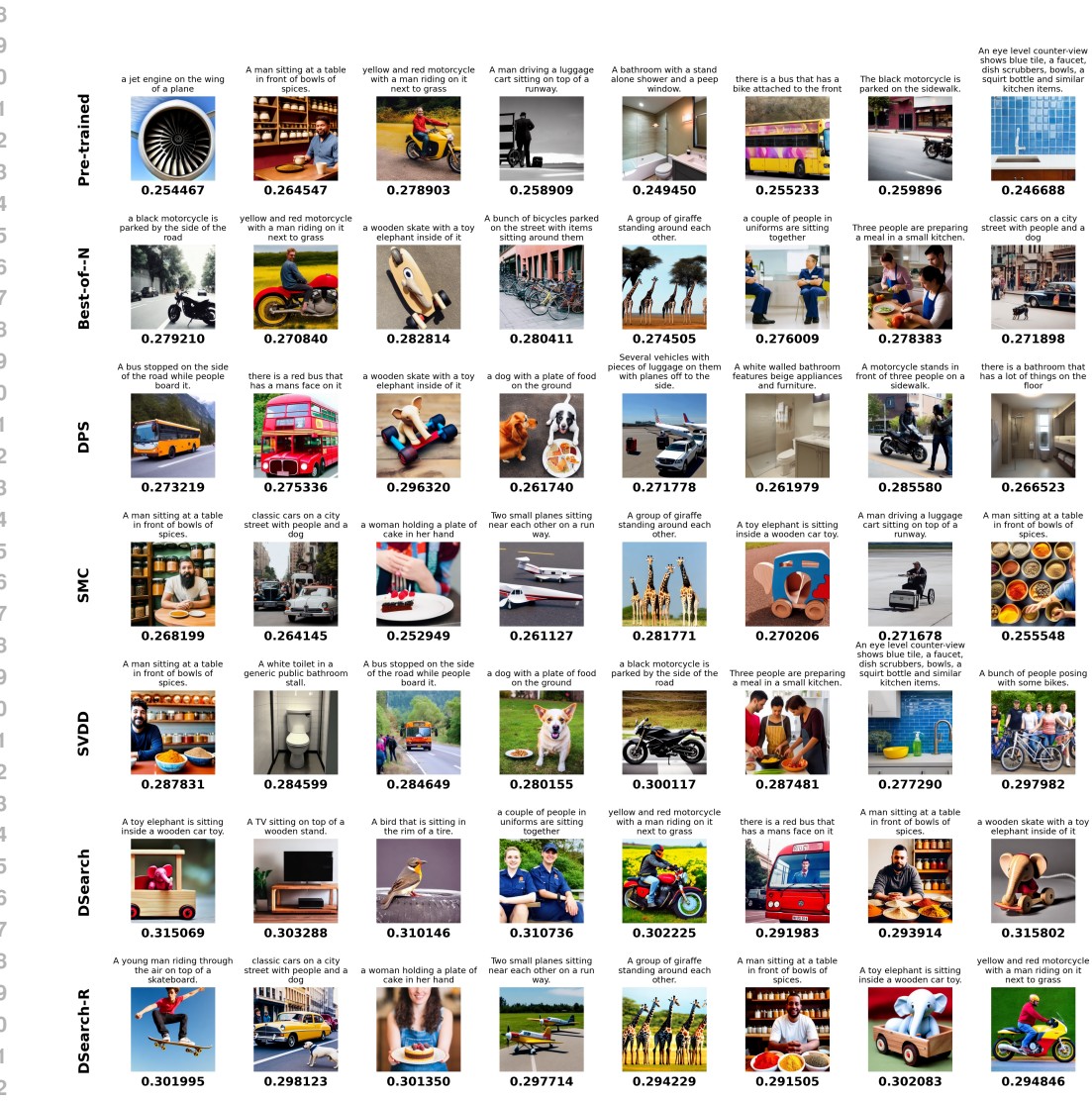

Figure 27: Visualization of generated images using different methods optimizing the reward of human preference score.

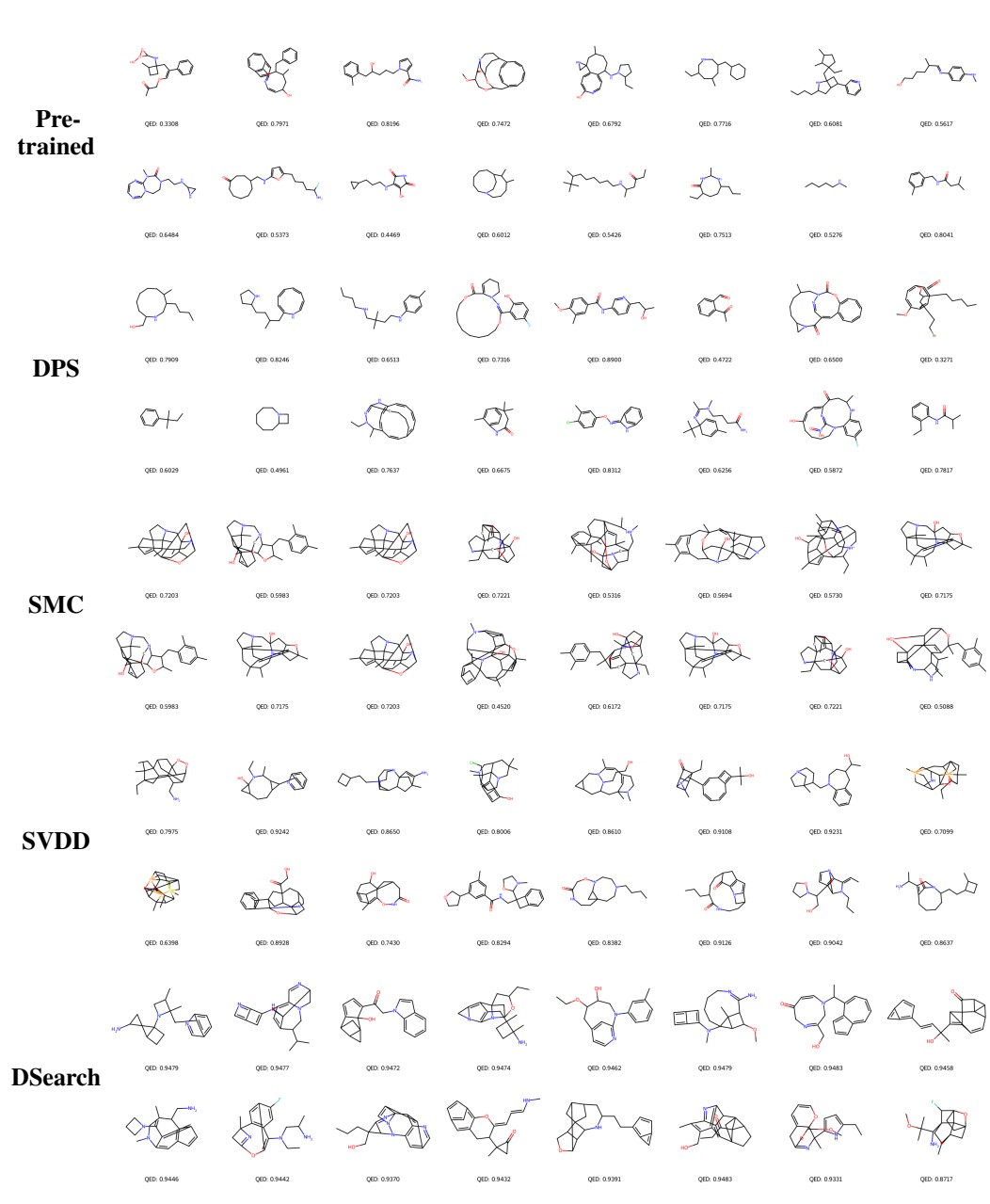

Figure 28: Visualization of generated molecules using different methods for optimizing QED.

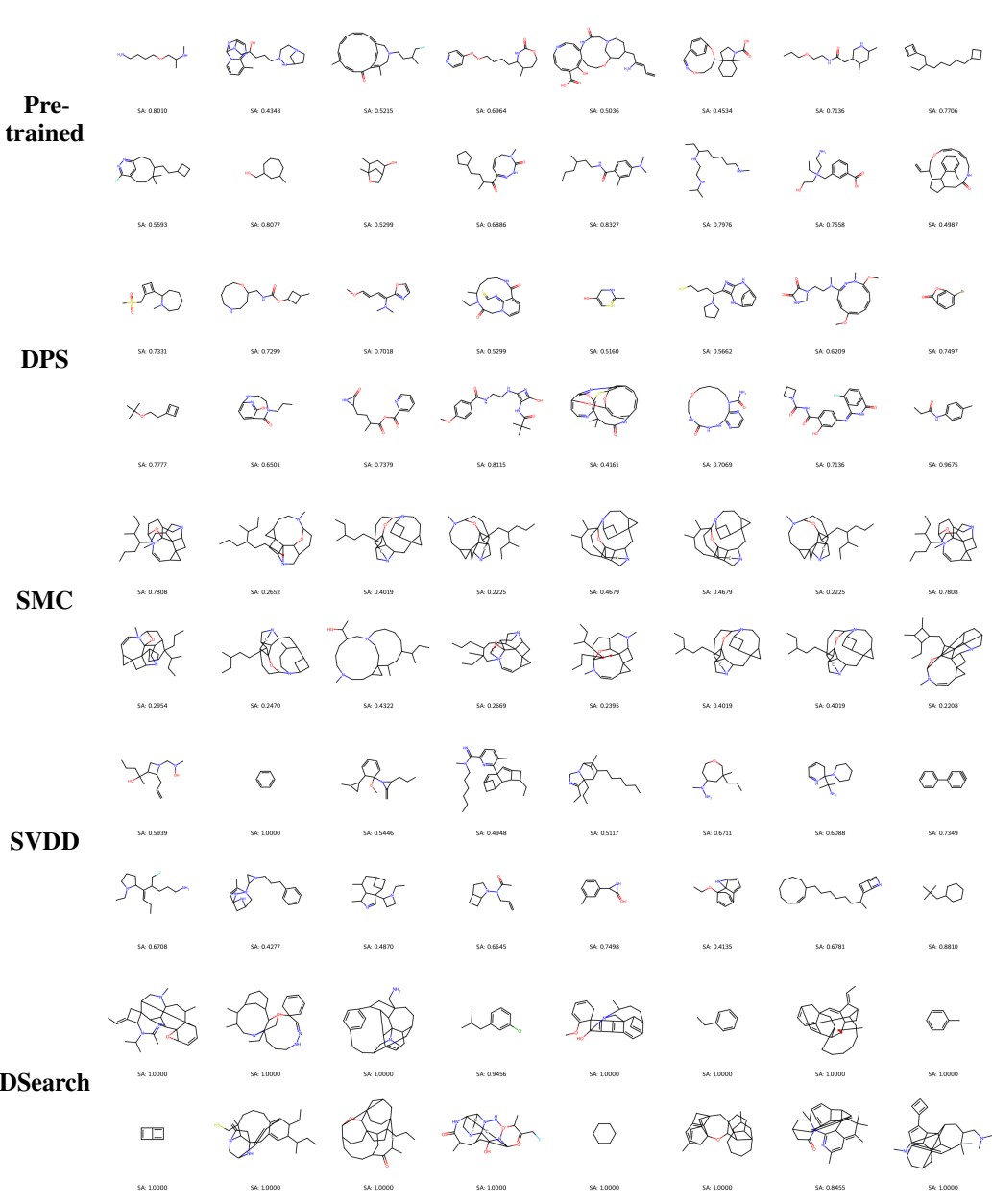

Figure 29: Visualization of generated molecules using different methods for optimizing SA.

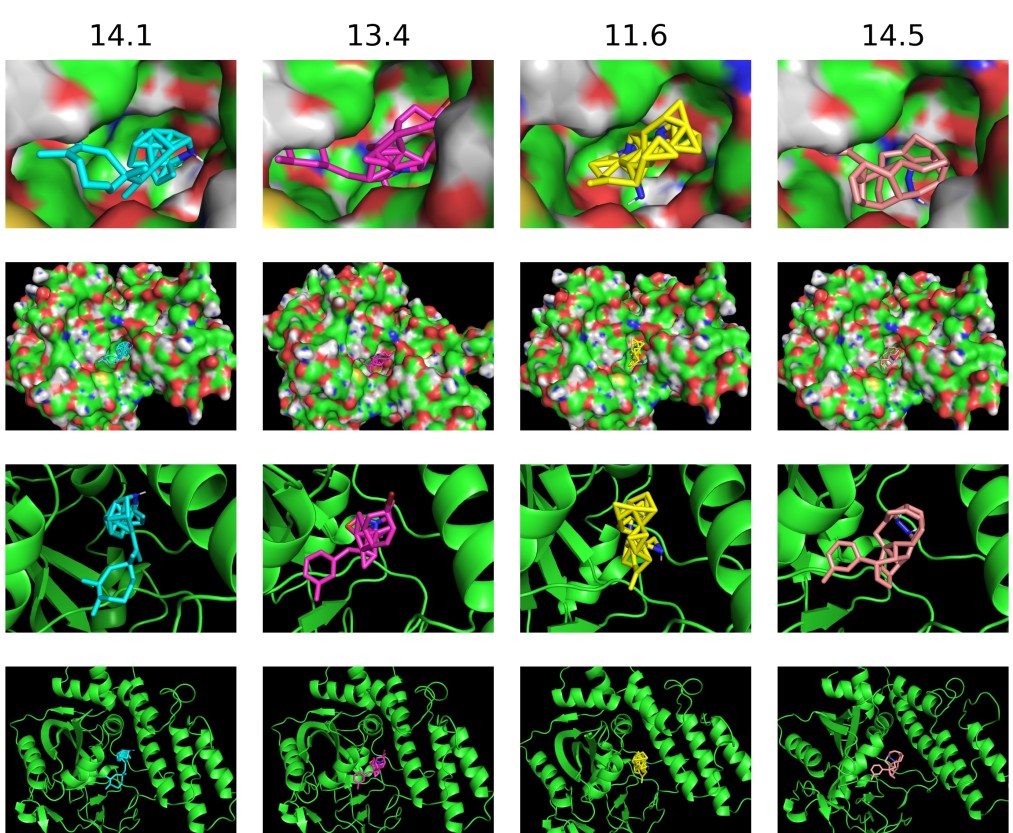

Figure 30: Visualization of generated molecules using DSearch optimizing the reward of docking score for parp1 (normalized as $max(-DS, 0)$).

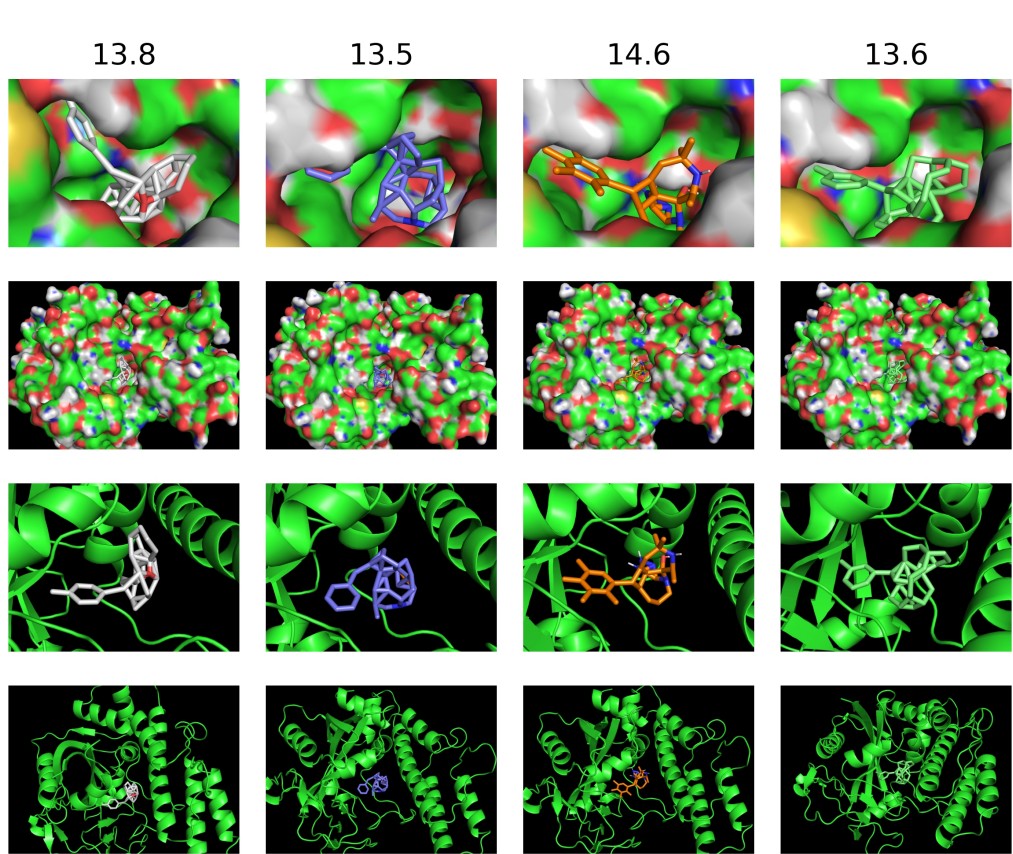

Figure 31: Visualization of more generated molecules using DSearch optimizing the reward of docking score for parp1 (normalized as $max(-DS, 0)$).

