# OpenReview forum: "Dynamic Search for Inference-Time Alignment in Diffusion Models"
_ICLR.cc/2026/Conference — Submitted to ICLR 2026_

### Official Review · Reviewer_pjyb · 2025-10-14

**Soundness:** 3
**Presentation:** 1
**Contribution:** 2
**Rating:** 4
**Confidence:** 3

**Summary:**

This paper designs a search algorithm that uses a pre-trained diffusion model as a prior distribution and targets non-differentiable reward functions. This method can be applied in domains such as images, biology, and chemistry to generate structures that meet requirements. Compared to existing methods such as classifier guidance, this approach does not require the reward function to be differentiable, expanding its application domains.

**Strengths:**

The look-ahead method proposed in this paper can obtain more accurate reward estimates when the diffusion model has not yet converged. In such optimization tasks, we generally consider the main cost to be fitness evaluation. Therefore, this technique can be regarded as relatively low-cost.

In terms of performance, if we assume the experiments are rigorous (I will explain in the weaknesses section why I say this), the improvements are still very significant.

**Weaknesses:**

The most confusing aspect of the paper, and perhaps its biggest flaw, lies in whether the experimental comparisons controlled for variables. In Figure 3, the prompts in (a) are not unified—DSearch and Pre-trained use different prompts, making it impossible to compare their performance. Figures (b, c) have the same problem. For images, the prompt condition has an enormous impact, and this variable needs to be properly controlled. Meanwhile, (d, e, f) do not compare against any other methods. If comparison is not possible, please state this clearly. Therefore, for the results in Table 1, I would like to understand the more specific measurement methods, including whether prompt variables were controlled.

Second, the computation budget comparison is not fair. In the DSearch method, the total number of value function evaluations can be up to $TC$, while the Best-of-N method only has $C$. In many cases, value function evaluation is the most time-consuming step, whereas the denoising steps can often be reduced by various methods. Therefore, I wonder whether Best-of-N could outperform DSearch if it were allowed $TC$ total value evaluations.

Third, the paper does not adequately compare with evolutionary algorithms. This article is essentially using a diffusion model as a prior distribution for search and optimization. If you strip away the diffusion model exterior, the interior is actually a type of evolutionary algorithm. However, the authors have not sufficiently cited literature in this area.

Although the authors' tasks all involve non-differentiable reward functions, if you reference evolutionary algorithms, you will find that many works approximate gradient estimation and derive beneficial insights from it. For example, OpenAI's OpenES uses random perturbations to estimate gradients, allowing the Adam optimizer to directly optimize non-differentiable tasks; Diffusion Evolution directly analogizes diffusion models and simultaneously estimates gradients for all individuals in the entire population on non-differentiable problems. So, although the authors uses search and selection as its narrative, the diffusion model leads to each step being a small update, which has hidden connections to gradient-base methods. If the paper could explore the connection between this method and gradient estimation and evolutionary algorithms, it would be greatly beneficial.

There are some minor issues elsewhere, including:

- Figure 4 appears to be a screenshot, which is unprofessional, and the "g" character in "Budget" is truncated. I recommend enlarging the text and converting it to a vector graphic.
- Line 147: Is the subscript of the first $t=1$ incorrect? Should the subsequent distribution be changed to $p(x_{t-1}|x_t)$? This does not seem to match the formula on line 221.
- Line 190: Although I can guess what $\left<s\right>$ is, I still recommend that you clarify the notation and unify the symbols.
- Line 1075: Is "$rr$" a typo?

**Questions:**

As I mentioned above in the Weaknesses, how exactly you compare the performance? And what variables are you controlling?

---

### Official Review · Reviewer_im5n · 2025-10-29

**Soundness:** 3
**Presentation:** 3
**Contribution:** 2
**Rating:** 4
**Confidence:** 4

**Summary:**

The paper proposes a heuristic inference-time searching method based on the beam search. The key contributions include (1) a look-ahead value estimation mechanism designed to reduce variance in value predictions, and (2) a dynamic scheduler that adjusts the beam width adaptively during search.

**Strengths:**

The authors have conducted extensive and rigorous experiments, covering a wide range of domains—from image data to biological datasets. The proposed method has been validated under multiple reward functions, demonstrating both robustness and general effectiveness.

**Weaknesses:**

While the approach is empirically solid, its technical novelty appears to be incremental. The method primarily combines heuristic improvements on classical beam search with a variance-reduced value estimation technique. The theoretical contribution remains somewhat limited, and the paper would benefit from a deeper analysis of why the proposed look-ahead estimation performs better beyond empirical observations.

There is an absence of a comprehensive description of the implementation details regarding the acquisition of the k look-ahead samples. In my understanding, this necessitates additional model inference procedures that depend on K. The paper lacks a detailed analysis of the additional computational burden incurred due to the variation of K. This is particularly evident when employing an extremely large diffusion model. And, in SVDD-PE, only one estimation is carried out for each sample at each step. In contrast, in Algorithm 1, an average of M estimations is performed per sample. Consequently, for the same (\bar{C}), DSearch necessitates (M\bar{C}) evaluations of the reward function, while SVDD-PE only requires (\bar{C}) evaluations. This should represent a significant additional computational cost, but based on the clock time provided by the authors, the Dsearch method is actually more efficient than SVDD. The paper lacks a theoretically more detailed computational analysis to better suit the analysis of computational and memory usage across multiple diffusion models and reward functions.

**Questions:**

Could the authors include results for SVDD with look-ahead value estimation? This would help isolate and highlight the contribution of the look-ahead value estimation.

In Figure 6, the label “DDES” should be corrected.

---

### Official Review · Reviewer_dQkp · 2025-11-01

**Soundness:** 2
**Presentation:** 3
**Contribution:** 2
**Rating:** 4
**Confidence:** 4

**Summary:**

This paper proposes DSearch, an inference-time alignment method that enables guidance of diffusion models with non-differentiable reward functions. DSearch casts sampling as tree search, dynamically adjusting expansion by noise level and a lookahead heuristic. As noise decreases, the schedule shrinks the beam size while increasing tree width. Nodes within each beam are greedily selected using a Monte Carlo lookahead with $K$-step rollouts and a one-step estimation. By keeping the product of beam size and tree width constant, DSearch maintains efficiency under a linear compute budget. Experiments across multiple domains show promising performance and effective reward-driven guidance.

**Strengths:**

- The authors provide an interesting tree-search framework of steering diffusion models for general black-box reward functions.
- The dynamic scheduling of beam size and tree width is novel and proven efficient.
- This paper is in general well-written with and proposed method is well-explained.

**Weaknesses:**

- A related work TreeG[1] explored a similar idea of using inference-time tree search for diffusion models, while it is not discussed or compared to but only cited as an "SMC based" method in related work. I find this unacceptable because it undermines the contribution and originality of this paper.
- While I understand that this paper treats reward functions as a black-box oracle, the HPS and aesthetic scores of images are differentiable, meaning that gradient-based diffusion sampling algorithms are applicable to these tasks. This makes it strange to only compare against gradient-free methods. Moreover, for the experiments with discrete diffusion, I would recommend including recent steering methods (e.g., [2,3]) as baselines too.



[1] Guo et al. "Training-Free Guidance Beyond Differentiability: Scalable Path Steering with Tree Search in Diffusion and Flow Models." NeurIPS 2025.

[2] Singhal et al. "A general framework for inference-time scaling and steering of diffusion models." ICML 2025.

[3] Chu et al. "Split Gibbs Discrete Diffusion Posterior Sampling." NeurIPS 2025.

**Questions:**

- When computing the heuristic function, the authors propose to sample $K$ steps ahead and estimate the reward by Tweedie's (one-step) estimation. Why is this approach considered more effective than a naive $K+1$ step sampler that evenly divides $[0, t]$?

---

### Official Review · Reviewer_tXsE · 2025-11-04

**Soundness:** 2
**Presentation:** 2
**Contribution:** 2
**Rating:** 4
**Confidence:** 4

**Summary:**

The paper proposes DSearch, an inference-time alignment method for diffusion models that reframes reward-guided generation as dynamic search over the denoising trajectory. Concretely, the method (i) fixes a per-step tree width and beam width under a compute budget; (ii) schedules expansion/pruning via “search scheduling” (subset of timesteps) and beam scheduling; and (iii) uses a look-ahead heuristic to score intermediate nodes. Experiments under matched compute across image, molecule, and bio-sequence tasks show that DSearch performs competitively with existing methods, and ablation studies are provided to understand various design choices.

**Strengths:**

- **Clear compute accounting & knobs.** The paper emphasizes fixed-budget comparisons and exposes intuitive controls (beam/tree widths and schedules).
- **Broad task coverage.** Experiments span images, DNA/RNA, and molecules, with naturalness/diversity metrics beyond reward (NLL, BRISQUE/CLIP-diversity, Tanimoto, etc.), which is good practice for reward-guided generation.
- **Scheduling ablations.** The paper studies several search/beam schedules and argues that exponential reduction of beam width with late-step focus is beneficial with corresponding  ablations.
- **Look-ahead heuristic explored.** The authors go beyond the 1-step posterior-mean proxy which is common in literature, in an effort to reduce the bias of these soft-value estimates.

**Weaknesses:**

My main concerns are around the positioning of the paper wrt prior work and some statements that need reconsideration.

- **Novelty claims vs prior search work in diffusion.** The paper positions itself as the first to frame inference-time alignment of diffusion models as a search problem (Sections 1 and 5). These statements risk overstating novelty, since plenty of existing works have done the same [1,2,3]. DTS [1] frames this problem as MCTS-style search by explicitly maintaining a tree and backtracking. TreeG [2] proposes a tree search over active paths and parallel exploration; SoP [3] performs a DFS-like search by repeated noising and denoising. Section 5 groups [2,3] as SMC-based methods which is incorrect, please (1) fix the description of [2,3] in related works section, (2) add discussion on other tree search methods [1], and (3) the novelty narrative should be tempered to clearly delineate what is new (e.g., the specific budgeted tree search over active paths and parallel exploration) versus what has been explored (search over denoising paths).
- **Tree search framing should be changed to just beam search.** The paper positions DSearch as a “tree-search” method throughout the paper, but the actual algorithm picks the best child per beam with a greedy selection rule, and then prunes this set to $b(t-1)$. There is no backtracking, no retained siblings for deeper reconsideration, and no explicit tree data structure beyond the current frontier. Functionally, this is a beam search with time-varying branching and beam sizes, or alternatively a beam search version of SVDD (that generates multiple children per node and picks the single best child for the next step; DSearch instead maintains multiple candidates and picks the best child for each such candidate).
- **Approximate soft-value derivation is mathematically loose.** The core heuristic $\hat{v}_t(x_t) = r(\hat{x}_0(x_t))$ is derived by replacing the conditional *distribution* $p^\text{pre}_0(\cdot | x_t)$ with the Dirac delta at the posterior mean $\delta(E[x_0|x_t])$ (Eq. 4, step (A)). I don’t see how this is mathematically valid; the pre-trained model distribution can be arbitrary, and collapsing it into a single point is not a reasonable approximation. The standard way of justifying the soft-value approximation is by applying Jensen’s inequality to the soft-value expression, followed by Tweedie’s formula to estimate the posterior mean (see [1]).
- **DSearch does not sample from the reward-tilted posterior in Eq. 1.** DSearch is largely a heuristic search method that obtains high-reward samples, but the writing in Section 2.2 gives the impression that DSearch aims to sample approximately from the target distribution.
- **Diversity metrics seem unreliable.** While the diversity metrics seem reasonable on paper, for several settings, the diversity of the pre-trained model is *worse* than the diversity of guided methods like DPS or DSearch. This is very surprising and indicates some sort of metric hacking, since samples from an unguided pre-trained model should always be more diverse than the narrow set of high-reward samples generated by other methods.
- **Figure 5 conclusions are statistically weak.** The distributions across different search/beam schedules in Figure 5 are very similar, and the text draws strong conclusions (e.g., exponential schedule is best). Without confidence intervals and multiple seeds, these differences look within noise. Please add multiple-seed statistics and formal tests.

*[1] Jain, Vineet, et al. "Diffusion Tree Sampling: Scalable inference-time alignment of diffusion models." The Thirty-ninth Annual Conference on Neural Information Processing Systems, 2025.*

*[2] Guo, Yingqing, et al. "Training-free guidance beyond differentiability: Scalable path steering with tree search in diffusion and flow models." arXiv preprint arXiv:2502.11420 (2025).*

*[3] Ma, Nanye, et al. "Inference-time scaling for diffusion models beyond scaling denoising steps." arXiv preprint arXiv:2501.09732 (2025).*

**Questions:**

- Can the authors specify the exact settings, like compute budget $\bar{C}$, look-ahead steps $K$, beam schedule etc., that were used for the set of results in Table 1?
- The scores in Table 1 and Figure 4 do not seem to match. Could the authors explain why these results are different?
- The authors refer to SVDD as “importance-sampling at each step” method, which is technically true, but the SVDD authors recommend using greedy selection ($\alpha=0$). What setting was used as a baseline in this paper?
- Given the similarity with SVDD, can the authors formalize a scenario where DSearch is *provably* better than SVDD (e.g., multi-modal intermediate posteriors where breadth across parents helps), or provide a counterexample where they are equivalent? A simple synthetic setting could clarify this.

---

### Meta-Review · Area_Chair_q62B · 2026-01-04

**Summary:**

The reviewers acknowledge that the proposed reward-optimization method is reasonable and technically sound. Specifically, they highlight the broad range of tasks and, aside from some concerns, are convincing. However, the reviewers raise several concerns regarding the proposed methodology and its evaluation. In particular, (1) the method is very closely related to the existing literature on using search algorithms and evolutionary algorithms to reward optimization, and (2) the paper does not provide any theoretical analysis for the procedure (e.g. the proof that it samples from a specific density), leaving it as a reasonable heuristic. Finally, the empirical evaluation still lacks comparisons with existing methods and raises concerns about metric choice and computational budget.

**Reviewer Concerns:**

The rebuttal was not submitted. Thus, no concerns have been addressed.

**Reviewer Scores:**

The reviewers could only reduce their scores since no rebuttal was submitted.

---

### Decision · Program_Chairs · 2026-01-26

Reject